# DGNet: Discrete Green Networks for Data-Efficient Learning of Spatiotemporal PDEs

**Yingjie Tan**
Qiuzhen College
Tsinghua University
`tanyj23@mails.tsinghua.edu.cn`

**Quanming Yao**
Department of Electronic Engineering
Tsinghua University
`qyaoaa@tsinghua.edu.cn`

**Yaqing Wang**[*]
Beijing Institute of Mathematical Sciences and Applications
`wangyaqing@bimsa.cn`

## Abstract

Spatiotemporal partial differential equations (PDEs) underpin a wide range of scientific and engineering applications. Neural PDE solvers offer a promising alternative to classical numerical methods. However, existing approaches typically require large numbers of training trajectories, while high-fidelity PDE data are expensive to generate. Under limited data, their performance degrades substantially, highlighting their low data efficiency. A key reason is that PDE dynamics embody strong structural inductive biases that are not explicitly encoded in neural architectures, forcing models to learn fundamental physical structure from data. A particularly salient manifestation of this inefficiency is poor generalization to *unseen source terms*. In this work, we revisit Green's function theory—a cornerstone of PDE theory—as a principled source of structural inductive bias for PDE learning. Based on this insight, we propose DGNet, a discrete Green network for data-efficient learning of spatiotemporal PDEs. The key idea is to transform the Green's function into a graph-based discrete formulation, and embed the superposition principle into the hybrid physics–neural architecture, which reduces the burden of learning physical priors from data, thereby improving sample efficiency. Across diverse spatiotemporal PDE scenarios, DGNet consistently achieves state-of-the-art accuracy using only tens of training trajectories. Moreover, it exhibits robust zero-shot generalization to unseen source terms, serving as a stress test that highlights its data-efficient structural design.

## 1 Introduction

Spatiotemporal partial differential equations (PDEs) form the foundation of modeling dynamical systems across science and engineering, governing phenomena in fluid dynamics (Hirsch, 2007), weather forecasting (Lynch, 2008), molecular dynamics (Lelievre & Stoltz, 2016), and energy systems (Ríos-Mercado & Borraz-Sánchez, 2015). Accurately solving such PDEs is crucial for scientific discovery and engineering design. Classical numerical solvers (Anderson, 2002; Evans et al., 2012) can provide reliable solutions but often become computationally prohibitive for large-scale irregular domains. Neural PDE solvers have therefore emerged as a promising alternative, aiming to approximate solution operators directly from data (Raissi et al., 2019; Brandstetter et al., 2022; Zeng et al., 2025). In practice, however, high-fidelity PDE data—whether collected from real systems or generated through simulation—are expensive to obtain. Existing neural PDE solvers typically require large numbers of training trajectories to achieve strong performance across the solution domain. Under limited data regimes, their accuracy degrades substantially, indicating low data efficiency. As a result, improving the data efficiency of neural PDE learning remains a critical yet underexplored challenge.

---

[*]Correspondence author

A fundamental reason for this limitation lies in the structural nature of PDE dynamics. PDE systems embody strong physical and mathematical inductive biases—such as locality, conservation laws, and superposition—that govern how states evolve over space and time. Yet most neural architectures do not explicitly encode these structural priors (Li et al., 2021; Lu et al., 2021; Sanchez-Gonzalez et al., 2020b), or only incorporate them weakly (Sanchez-Gonzalez et al., 2020a; Horie & Mitsume, 2022; Bishnoi et al., 2024). Consequently, models must learn the underlying operator structure and physical constraints directly from data, leading to high sample complexity and reduced data efficiency. This issue becomes particularly pronounced in the presence of source terms $f(\mathbf{x}, t)$, which represent external forcing applied over space and time. Examples include time-varying heat sources in conduction (Hahn & Özisik, 2012), body forces in fluid dynamics (Pope, 2001), time-dependent currents in electromagnetics (Taflove et al., 2005). Training trajectories typically cover only a subset of possible source patterns, existing solvers often struggle to extrapolate to unseen source patterns. Poor generalization to unseen source terms thus serves as a representative manifestation of insufficient inductive bias and data inefficiency in current neural PDE approaches.

To improve data efficiency, it is therefore essential to introduce stronger structural priors that meaningfully reduce the learning space and exploration burden of neural solvers. Motivated by this perspective, we revisit Green's function theory, a foundational principle in PDE analysis. Green's function provides a principled foundation, it effectively extracts the solution $u$ from the direct action of the operator $\mathcal{L}_{\mathbf{x}}$, recasting the PDE solution into two explicit components—the homogeneous evolution and the forced response. This decomposition encodes the superposition structure intrinsic to many PDE systems, which provides a principled form of inductive bias that reduces the need to rediscover fundamental behavior from data. Moreover, this formulation naturally separates system evolution from source responses, offering a principled foundation for improved generalization under varying source conditions.

Building on this insight, we develop DGNet for spatiotemporal PDE learning on irregular domains commonly encountered in scientific and engineering applications. The core idea is to construct a discrete Green's formulation on graphs that preserves the superposition structure in a computable update rule. By embedding this discrete operator into a hybrid physics–neural architecture, we combine principled physical priors with learnable corrections from graph neural networks (GNNs). This design reduces the burden of learning fundamental dynamics from data while retaining flexibility on complex geometries. Empirically, DGNet demonstrates strong data efficiency and robust generalization to unseen source terms.

Our contributions can be summarized as follows:

- We attribute the limited data efficiency of neural PDE solvers to insufficient structural inductive bias. We provide the insight that Green's function theory offers a principled prior, which can substantially reduce the need to rediscover fundamental physical structure from data.

- We introduce DGNet, a discrete Green network. By explicitly embedding the superposition principle into the hybrid physics–neural architecture, DGNet encodes physical structure directly in the model rather than requiring it to be rediscovered from data, thereby promoting data-efficient learning by design.

- Empirically, DGNet achieves state-of-the-art accuracy across diverse spatiotemporal PDE systems using only tens of training trajectories, highlighting its data-efficient structural design. Furthermore, it exhibits robust zero-shot generalization to unseen source terms.

## 2 RELATED WORK

**Physics-informed neural networks (PINNs).** PINNs (Raissi et al., 2019) incorporate PDE residuals and boundary conditions as soft constraints in the training loss, enabling models to fit solutions at sparsely sampled collocation points. However, PINNs essentially learn a specific solution under a fixed setup, and thus cannot generalize to domains, parameters, or forcing terms beyond the training distribution. Variants (Yu et al., 2022; Sukumar & Srivastava, 2022; Costabal et al., 2024) have attempted to improve stability or computational efficiency, yet data requirements grow rapidly when accurate solutions are required across the solution domain.

**Neural operators.** Neural operator methods directly learn mappings between infinite-dimensional function spaces from data. Representative works include DeepONet (Lu et al., 2021) and the Fourier

Neural Operator (FNO) (Li et al., 2021), along with many extensions in the spectral or kernel domains (Guibas et al., 2021; Li et al., 2023; George et al., 2024; Tran et al., 2023). These models achieve generalization across resolutions and physical parameters. However, they typically incorporate limited physical structure into the model, and learning operator mappings in function spaces requires substantial amounts of data.

**Graph-based PDE solvers.** For spatiotemporal PDEs on irregular domains, graphs provide a natural backbone. The Graph Network Simulator (GNS) (Sanchez-Gonzalez et al., 2020b) introduced the encoder–processor–decoder framework for physical simulation. Subsequent works enhanced physical consistency by embedding conservation laws (Sanchez-Gonzalez et al., 2020a; Cranmer et al., 2020; Bishnoi et al., 2023), incorporating operator blocks (Seo et al., 2019; Horie & Mitsume, 2022; Zeng et al., 2025), or enforcing invariants such as momentum conservation (Bishnoi et al., 2024). BENO (Wang et al., 2024) further drew inspiration from Green's functions, but focused on time-independent elliptic PDEs. Despite the incorporation of various inductive biases, existing graph-based solvers lack a principled structural foundation, still rely on substantial data to learn complex physical dynamics.

**Generalization challenges in PDE solvers.** Generalization is central to neural PDE solvers. Neural operators such as FNO (Li et al., 2021) generalize across resolutions, DeepONet (Lu et al., 2021) across parameters, and BroGNet (Bishnoi et al., 2024) across system sizes. Yet a critical gap remains: none of these approaches can generalize to unseen source terms, which frequently arise in scientific and engineering systems. Such varying sources are ubiquitous in practice, including time-varying heat sources in conduction (Hahn & Özisik, 2012), body forces in fluid dynamics (Pope, 2001), time-dependent currents in electromagnetics (Taflove et al., 2005), and seismic excitations in elasticity (Aki & Richards, 2002).

## 3 DGNET: DISCRETE GREEN NETWORKS

In this section we introduce DGNet, our proposed model for data-efficient spatiotemporal PDE learning enabled by structural inductive bias. We begin in Section 3.1 by formalizing the problem setup. In Section 3.2, we introduce Green's function and the superposition principle, which naturally decompose the PDE solution into state evolution and source response. In Section 3.3, we derive the discrete Green formulation by discretizing both space and time, leading to an update rule that preserves the superposition structure. In Section 3.4, we describe our physics–neural hybrid operator that combines numerical discretization with a GNN-based correction to construct a high-fidelity solver. Finally, Section 3.5 describes the prediction and training procedure.

### 3.1 PROBLEM FORMULATION

We consider physical systems governed by spatiotemporal partial differential equations (PDEs) as

$$\frac{\partial u(\mathbf{x}, t)}{\partial t} = \mathcal{L}_{\mathbf{x}}[u(\mathbf{x}, t)] + f(\mathbf{x}, t), \quad (\mathbf{x}, t) \in \Omega \times (0, T], \tag{1}$$

where $u(\mathbf{x}, t) \in \mathbb{R}$ denotes the physical state, $\mathcal{L}_{\mathbf{x}}$ is a spatial differential operator characterizing the system dynamics, and $f(\mathbf{x}, t)$ is the source term.

The evolution is uniquely determined by an initial condition (IC) and boundary conditions (BCs). The IC specifies $u(\mathbf{x}, 0) = u_0(\mathbf{x})$. On the boundary $\partial\Omega$, we primarily consider Dirichlet BCs $u(\mathbf{x}, t) = g(\mathbf{x}, t)$ for $\mathbf{x} \in \partial\Omega_D$ and Neumann BCs $\frac{\partial u(\mathbf{x}, t)}{\partial \mathbf{n}} = h(\mathbf{x}, t)$ for $\mathbf{x} \in \partial\Omega_N$, with $\partial\Omega_D \cup \partial\Omega_N = \partial\Omega$ and $\partial\Omega_D \cap \partial\Omega_N = \emptyset$. Other BC types (e.g., Robin, periodic) are accommodated in the same framework.

In many real-world systems, the source $f(\mathbf{x}, t)$ varies substantially across space and time and thus plays a decisive role in the solution behavior. In practice, models are typically trained on only a limited subset of possible source patterns, and their performance often degrades when confronted with unseen sources. Generalization to varying source terms serves as a stringent test of data efficiency and structural inductive bias in neural PDE solvers.

## 3.2 Green's Function Representation

Directly solving the PDE (1) is often challenging. The Green's function method from PDE theory (Arfken et al., 2013) provides a principled way to characterize the system response. The Green's function $G(\mathbf{x}, t; \mathbf{x}', \tau)$ is uniquely determined by

$$\left( \frac{\partial}{\partial t} - \mathcal{L}_{\mathbf{x}} \right) G(\mathbf{x}, t; \mathbf{x}', \tau) = \delta(\mathbf{x} - \mathbf{x}') \, \delta(t - \tau), \tag{2}$$

where $\delta(\cdot)$ is the Dirac delta function. Physically, $G(\mathbf{x}, t; \mathbf{x}', \tau)$ describes the influence observed at $(\mathbf{x}, t)$ due to a unit-strength point source applied at $(\mathbf{x}', \tau)$.

Crucially, the Green's function acts as a mathematical device to "extract" the solution $u$ from the direct action of the operator $\mathcal{L}_{\mathbf{x}}$. Instead of learning $\mathcal{L}_{\mathbf{x}}[u]$ as an inseparable whole, the Green representation rewrites the PDE solution in terms of a propagation kernel defined by $\mathcal{L}_{\mathbf{x}}$ and a convolution with the source term. According to the superposition principle (Boyce et al., 2017), the complete solution $u(\mathbf{x}, t)$ can therefore be expressed as

$$u(\mathbf{x}, t) = \underbrace{\int_{\Omega} G(\mathbf{x}, t; \mathbf{x}', 0) \, u_0(\mathbf{x}') \, d\mathbf{x}'}_{\text{Evolution of initial state}} + \underbrace{\int_0^t \int_{\Omega} G(\mathbf{x}, t; \mathbf{x}', \tau) \, f(\mathbf{x}', \tau) \, d\mathbf{x}' d\tau}_{\text{Response to source term}}. \tag{3}$$

This representation makes two key points explicit. First, the solution naturally decomposes into the evolution of the initial condition and the accumulated response to the source term. Second, it enables a unified treatment of diverse and time-varying forcing conditions. In the next section, we describe how this continuous formulation is discretized on graphs, leading to a discrete Green's function that forms the foundation of our learning architecture and provides a strong structural inductive bias for data-efficient learning.

## 3.3 Discretization

While Green's function offers an elegant continuous representation, it is seldom available in closed form for complex domains, and direct evaluation of the associated integrals is computationally prohibitive. We therefore derive a discrete Green formulation on graphs that preserves the superposition structure within a tractable update rule. In this discrete setting, the Green's function naturally acts as a propagation operator, evolving the system state while incorporating source effects at each time step.

**Graph-based ODE system.** Graph-based discretization is particularly suitable for spatiotemporal PDEs on irregular meshes, as it provides a flexible representation of spatial operators and naturally supports message-passing based corrections. Therefore, in DGNet, we discretize the spatial domain $\Omega$ into $N$ nodes $\{\mathbf{x}_0, \ldots, \mathbf{x}_{N-1}\}$, and represent it as a graph $\mathcal{G} = (\mathcal{V}, \mathcal{E})$, where $\mathcal{V}$ is the set of nodes and $\mathcal{E}$ is the edge set constructed once using Delaunay triangulation. Each node $i \in \mathcal{V}$ corresponds to a spatial location $\mathbf{x}_i$.

At a discrete time $t^k$, the physical state values $\{u(\mathbf{x}_i, t^k)\}_{i=0}^{N-1}$ form a state vector $\mathbf{u}^k = \left( u(\mathbf{x}_0, t^k), \ldots, u(\mathbf{x}_{N-1}, t^k) \right) \in \mathbb{R}^N$, and similarly the source term values $\{f(\mathbf{x}_i, t^k)\}_{i=0}^{N-1}$ form a vector $\mathbf{f}^k = \left( f(\mathbf{x}_0, t^k), \ldots, f(\mathbf{x}_{N-1}, t^k) \right) \in \mathbb{R}^N$. When the time index $k$ is not critical, we simply denote them by $\mathbf{u}$ and $\mathbf{f}$. The continuous PDE (1) can then be written in discrete form as

$$\frac{d\mathbf{u}}{dt} = \mathbf{L}\mathbf{u} + \mathbf{f}, \tag{4}$$

where $\mathbf{L} \in \mathbb{R}^{N \times N}$ is the matrix discretization of the spatial operator $\mathcal{L}_{\mathbf{x}}$. For node $i$, the action of $\mathbf{L}$ on the state vector $\mathbf{u}$ is $[\mathbf{L}\mathbf{u}]_i = \sum_{j=1}^N [\mathbf{L}]_{ij} \, [\mathbf{u}]_j$, where $[\cdot]_{ij}$ denotes the $(i, j)$-th entry of a matrix and $[\cdot]_j$ denotes the $j$-th component of a vector.

**Time integration.** We apply a midpoint (Crank–Nicolson) discretization on $[t_k, t_{k+1}]$: let $\Delta t = t_{k+1} - t_k$, the time increment is $(\mathbf{u}^{k+1} - \mathbf{u}^k)/\Delta t$, and the right-hand side is approximated by a trapezoidal average, yielding

$$\frac{\mathbf{u}^{k+1} - \mathbf{u}^k}{\Delta t} = \tfrac{1}{2}\left( \mathbf{L}\mathbf{u}^k + \mathbf{L}\mathbf{u}^{k+1} \right) + \tfrac{1}{2}\left( \mathbf{f}^k + \mathbf{f}^{k+1} \right). \tag{5}$$

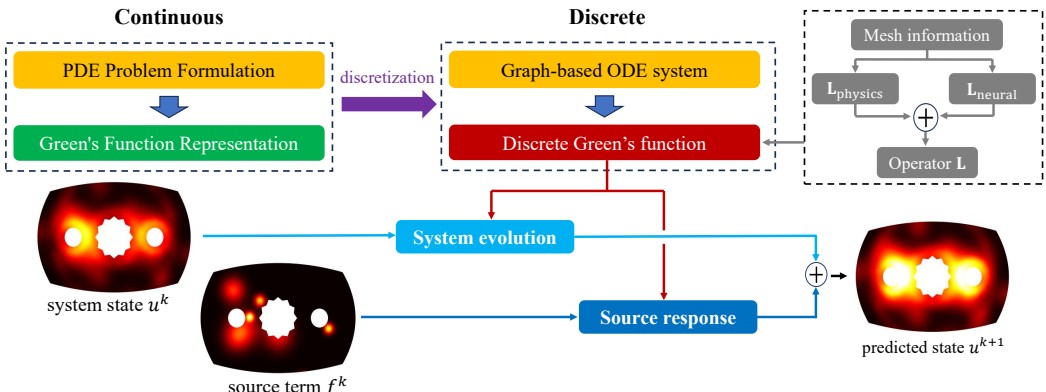

Figure 1: Overview of DGNet architecture. The model centers on a hybrid operator $\mathbf{L} = \mathbf{L}_{\text{physics}} + \mathbf{L}_{\text{neural}}$, where $\mathbf{L}_{\text{physics}}$ encodes gradient and Laplacian discretizations and $\mathbf{L}_{\text{neural}}$ is a GNN-based correction for mesh-induced errors. This operator is integrated into the discrete Green's function update (Eq. 7), which naturally combines system evolution with source-term response and provides structural inductive bias for data-efficient learning.

Rearranging terms yields a sparse linear system:

$$\left(\mathbf{I} - \tfrac{\Delta t}{2}\mathbf{L}\right)\mathbf{u}^{k+1} = \left(\mathbf{I} + \tfrac{\Delta t}{2}\mathbf{L}\right)\mathbf{u}^{k} + \tfrac{\Delta t}{2}\left(\mathbf{f}^{k} + \mathbf{f}^{k+1}\right). \tag{6}$$

**Discrete Green's function.** From Eq. 6, the discrete Green's function is identified as

$$\mathbf{G}(\Delta t) = \left(\mathbf{I} - \tfrac{\Delta t}{2}\mathbf{L}\right)^{-1}.$$

Thus the single-step update becomes

$$\mathbf{u}^{k+1} = \mathbf{G}(\Delta t)\left(\mathbf{I} + \tfrac{\Delta t}{2}\mathbf{L}\right)\mathbf{u}^{k} + \mathbf{G}(\Delta t)\tfrac{\Delta t}{2}\left(\mathbf{f}^{k} + \mathbf{f}^{k+1}\right). \tag{7}$$

This discrete formulation mirrors the superposition principle (3): the next state $\mathbf{u}^{k+1}$ results from the propagation of the current state under $\mathbf{G}(\Delta t)$ together with the accumulated response to the source term within the interval $\Delta t$. A full derivation of this update scheme is provided in Appendix A.

By embedding structural priors directly into the model, DGNet narrows the space of admissible solutions consistent with physical laws. This reduces the burden on data to recover solution operator and improves learning efficiency, particularly in limited-trajectory regimes.

Since naively computing the inverse $\left(\mathbf{I} - \tfrac{\Delta t}{2}\mathbf{L}\right)^{-1}$ is infeasible for large systems. Instead, we solve the equivalent sparse linear system, where $\mathbf{L}$ is already sparse due to local mesh interactions. Importantly, the coefficient matrix $\left(\mathbf{I} - \tfrac{\Delta t}{2}\mathbf{L}\right)$ depends only on the static mesh geometry and thus remains fixed throughout rollout. We therefore adopt a "factorize once, solve many times" strategy: a single sparse LU factorization is performed before rollout, and its factors are cached and reused for all time steps via efficient forward/backward substitution. This reduces per-step cost to nearly linear in the number of nonzeros, making the discrete Green solver practical for large meshes.

### 3.4 Physics–Neural Hybrid Operator

The core of our architecture lies in constructing a high-fidelity operator $\mathbf{L}$ for the discrete ODE system. Instead of relying solely on data-driven models or purely hand-crafted discretizations, we propose a hybrid approach that combines the best of both worlds:

$$\mathbf{L} = \mathbf{L}_{\text{physics}} + \mathbf{L}_{\text{neural}}. \tag{8}$$

Here, $\mathbf{L}_{\text{physics}}$ is built directly from mesh geometry using numerical discretization techniques, ensuring consistency with physical laws. Meanwhile, $\mathbf{L}_{\text{neural}}$ is a learnable correction term, parameterized by a GNN, that compensates for discretization errors on irregular meshes.

### 3.4.1 PHYSICS PRIOR OPERATOR $\mathbf{L}_{\text{PHYSICS}}$

Among the various spatial operators appearing in PDEs, the gradient and Laplacian are the most fundamental and widely used components (e.g., in diffusion, convection–diffusion, and Poisson-type equations). Accordingly, we construct $\mathbf{L}_{\text{physics}}$ directly from mesh geometry using established discretization schemes, namely the Green–Gauss theorem (Löhner, 2008) and the discrete Laplace–Beltrami operator (Meyer et al., 2003). These methods are widely used in computational physics and provide physically consistent priors on irregular meshes, ensuring that $\mathbf{L}_{\text{physics}}$ encodes reliable physics knowledge. Accordingly, $\mathbf{L}_{\text{physics}}$ is instantiated as follows:

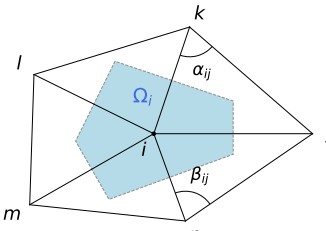

- **Gradient operator.** $[\mathbf{L}_{\text{physics}}]_{ij} = \frac{m_{ij}l_{ij}}{2|\Omega_i|}$ if $j \in \mathcal{N}(i)$ and, $[\mathbf{L}_{\text{physics}}]_{ii} = -\sum_{k \in \mathcal{N}(i)} \frac{m_{ik}l_{ik}}{2|\Omega_i|}$, other elements are zero. Where $|\Omega_i|$ is the control volume, and $m_{ij}$ is the scalar projection of unit normal $\mathbf{n}_{ij}$, $l_{ij}$ is the face length.

- **Laplacian operator.** $[\mathbf{L}_{\text{physics}}]_{ij} = \frac{w_{ij}}{|\Omega_i|}$ if $j \in \mathcal{N}(i)$ and $[\mathbf{L}_{\text{physics}}]_{ii} = -\frac{1}{|\Omega_i|}\sum_{k \in \mathcal{N}(i)} w_{ik}$, other elements are zero. With cotangent weights $w_{ij} = \frac{1}{2}(\cot \alpha_{ij} + \cot \beta_{ij})$ and Voronoi area $|\Omega_i|$.

Figure 2: Geometric variables for the discrete operator.

Where $\Omega_i$ is the control region of node $i$, $\mathcal{N}(i)$ is the first-order neighbor set of node $i$, $\alpha_{ij}$ and $\beta_{ij}$ are the two angles opposite to edge $(i, j)$. The geometric variables are illustrated in Figure 2

### 3.4.2 NEURAL CORRECTION OPERATOR $\mathbf{L}_{\text{NEURAL}}$

Although $\mathbf{L}_{\text{physics}}$ provides a principled foundation, it fundamentally approximates a continuous operator on a discretized mesh, where truncation errors are inherent and inevitable. Even though these errors may diminish on high-resolution or regular meshes, the gap between discrete and continuous dynamics persists. To bridge this gap and recover operator consistency without resorting to computationally expensive fine meshing, we introduce a correction matrix $\mathbf{L}_{\text{neural}}$, learned by a GNN following the Encode–Process–Decode paradigm (Sanchez-Gonzalez et al., 2020b).

**Encoder.** Node features consist of the spatial coordinates $\mathbf{x}_i$ and node type $C_i$ (interior or boundary), which are mapped to embeddings as $\mathbf{h}_i^0 = \text{NodeEncoder}([\mathbf{x}_i, C_i])$. Edge features consist of the relative displacement $\mathbf{x}_{ij} = \mathbf{x}_j - \mathbf{x}_i$ and distance $d_{ij} = \|\mathbf{x}_j - \mathbf{x}_i\|_2$, mapped to embeddings as $\mathbf{e}_{ij} = \text{EdgeEncoder}([\mathbf{x}_{ij}, d_{ij}])$.

**Processor.** The processor consists of $M$ layers of message-passing neural networks (MPNNs). At the $m$-th layer, we obtain

$$\mathbf{h}_i^{m+1} = \text{Update}\left(\mathbf{h}_i^m, \sum_{j \in \mathcal{N}(i)} \text{Message}(\mathbf{h}_i^m, \mathbf{h}_j^m, \mathbf{e}_{ij})\right) + \mathbf{h}_i^m. \tag{9}$$

**Decoder.** After $M$ layers, for each edge $(i, j)$, we concatenate final node embeddings and pass them through a decoder to predicts edge-level corrections:

$$[\mathbf{L}_{\text{neural}}]_{ij} = \text{Decoder}([\mathbf{h}_i^M, \mathbf{h}_j^M]). \tag{10}$$

For simplicity, we implement all neural components, including the node encoder, edge encoder, message function, update function, and decoder, as multilayer perceptrons (MLPs). Theoretical analysis of the consistency and stability properties of the learned correction operator is provided in Appendix E.

### 3.5 PREDICTION AND TRAINING

With the hybrid operator $\mathbf{L} = \mathbf{L}_{\text{physics}} + \mathbf{L}_{\text{neural}}$, the system state is updated by the discrete Green solver in Eq. 7, yielding predictions $\hat{\mathbf{u}}^k$ at each global time step $t_k$.

To further improve accuracy, we include a lightweight residual GNN correction module in the prediction path. This design follows prior physics–neural hybrid solvers (Meng & Karniadakis, 2020; Wu et al., 2024b; Long et al., 2023; Zeng et al., 2025), which demonstrate that small residual modules can effectively approximate dynamics not explicitly captured by the physical operator, while preserving interpretability of the operator-based formulation.

For training, the full trajectory $(\mathbf{u}^0, \ldots, \mathbf{u}^{T-1})$ is segmented into shorter sub-sequences of length $Q \ll T$: $(\mathbf{u}^{s_0}, \mathbf{u}^{s_1}, \ldots, \mathbf{u}^{s_{Q-1}})$, where $s_q$ denotes the global time index of the $q$-th step in the sub-sequence (corresponding to time point $t_{s_q}$). The model rolls out from $\mathbf{u}^{s_0}$ to predict $(\hat{\mathbf{u}}^{s_1}, \ldots, \hat{\mathbf{u}}^{s_{Q-1}})$. Following prior work (Brandstetter et al., 2022; Pfaff et al., 2021; Sanchez-Gonzalez et al., 2020b), we apply the pushforward trick and inject small noise into $\mathbf{u}^{s_0}$ during training to alleviate error accumulation and improve robustness. The loss is computed only on the first and last predictions of each sub-sequence:

$$\mathcal{L} = \|\hat{\mathbf{u}}^{s_1} - \mathbf{u}^{s_1}\|^2 + \|\hat{\mathbf{u}}^{s_{Q-1}} - \mathbf{u}^{s_{Q-1}}\|^2.$$

The learnable parameters are optimized with Adam using a decayed learning rate schedule.

## 4 EXPERIMENTS

In this section, we empirically evaluate DGNet across diverse scenarios to assess its data efficiency and generalization capability. We consider multiple spatiotemporal PDE systems and examine performance under a limited-trajectory training regime. Section 4.1 describes the experimental setup, including datasets, baselines, and evaluation metrics. Section 4.2 presents quantitative and qualitative results across different categories of PDE systems. Section 4.3 provides ablation studies to analyze the contribution of each architectural component. The source code is publicly available.[1]

### 4.1 EXPERIMENTAL SETUP

Table 1: Governing equations and physical contexts for experimental scenarios.

| Name | Physical Scenario | Mathematical Formulation | Meaning of Source Term |
|---|---|---|---|
| Allen–Cahn | Phase separation | $\partial_t u = \epsilon^2 \nabla^2 u - (u^3 - u)$ | Driving force for separation |
| Fisher–KPP | Population dynamics | $\partial_t u + \mathbf{c} \cdot \nabla u = \rho u(1-u)$ | Logistic growth |
| FitzHugh–Nagumo | Excitable systems | $\partial_t u = D_u \nabla^2 u + \Gamma(-u^3+u-v); \partial_t v = D_v \nabla^2 v + \Gamma\beta(u - \alpha v)$ | Excitation–recovery coupling |
| Contaminant Transport | Channel flow with obstacles | $\partial_t c + \mathbf{u} \cdot \nabla c = D\nabla^2 c + \rho_g c(1-c) - k_d c$ | Reaction (generation/decay) |
| Laser Heat | Laser heating | $\rho c_p \partial_t T = k\nabla^2 T - h(T - T_{\text{amb}}) + Q(x,t)$ | Moving laser heat source |

**Datasets.** We evaluate DGNet on three categories of spatiotemporal PDE systems: (1) *classical equations* (Allen–Cahn, Fisher–KPP, FitzHugh–Nagumo) to validate accuracy on canonical diffusion-, advection-, and reaction-driven dynamics, (2) *complex geometric domains* (contaminant transport with cylinder, sediment, and irregular obstacles) to test robustness on irregular meshes and boundary conditions, and (3) *generalization to unseen source terms* (laser heat treatment) to evaluate adaptability to novel forcing conditions. Table 1 summarizes the governing equations, physical contexts, and the meaning of the source term in each scenario. Detailed dataset parameters (domains, mesh sizes, time steps, boundary conditions, and trajectory splits) are deferred to Appendix B.

**Baselines.** We compare DGNet with representative neural PDE solvers: DeepONet (Lu et al., 2021), MGN (Pfaff et al., 2021), MP-PDE (Brandstetter et al., 2022), PhyMPGN (Zeng et al., 2025), and BENO (Wang et al., 2024). These cover operator-learning, graph-based, and hybrid paradigms. All baselines are tuned to have comparable numbers of parameters and training costs. Implementation details are provided in Appendix C.

**Evaluation Metrics.** We report Mean Squared Error (MSE) and Relative $\ell_2$ Error (RNE). RNE is defined as $\|\hat{u} - u\|_2 / \|u\|_2$, where $\hat{u}$ and $u$ denote predicted and ground-truth states. We report $\log_{10}$ MSE for readability, while the relative ranking remains consistent with raw MSE.

---
[1]https://github.com/tanyingjie01/DGNet

Notably, all experiments are conducted in a limited-data regime, with only tens of training trajectories per scenario. In contrast, neural PDE solvers are often trained with substantially larger datasets—ranging from hundreds to thousands of trajectories—to achieve stable performance. We evaluate under this low-data setting to highlight the data efficiency of DGNet.

## 4.2 PERFORMANCE COMPARISON

Table 2 summarizes results across all experimental scenarios, covering the three categories of spatiotemporal PDE systems considered in this paper. Even under the limited-trajectory training regime, DGNet achieves stable and state-of-the-art performance across all tasks and metrics, highlighting the data efficiency of its structural design. Further analysis of efficiency and scalability is provided in Appendix D, and additional experimental results on broader scenarios are presented in Appendix F.

Table 2: Results across scenarios from three categories of spatiotemporal PDE systems under the limited-trajectory training. For both MSE (in log scale) and RNE, lower values indicate better performance. The best results are highlighted in bold.

| Scenario | Metric | DeepONet | MGN | MP-PDE | BENO | PhyMPGN | DGNet |
|---|---|---|---|---|---|---|---|
| Allen-Cahn | MSE | 2.60e-01 | 2.70e-01 | 8.52e-01 | 2.52e+00 | 5.16e-01 | **8.75e-03** |
| | RNE | 0.6686 | 0.6813 | 1.2109 | 2.0813 | 0.9420 | **0.0188** |
| Fisher-KPP | MSE | 3.05e-02 | 3.66e-03 | 9.90e-02 | 6.26e-02 | 1.50e-02 | **2.59e-04** |
| | RNE | 0.4181 | 0.1448 | 0.7530 | 0.5989 | 0.9270 | **0.0238** |
| FitzHugh-Nagumo | MSE | 2.49e-06 | 3.75e-05 | 6.464e-06 | 2.14e-04 | 1.69e-03 | **1.18e-07** |
| | RNE | 0.9745 | 3.4815 | 1.4454 | 8.3106 | 23.5696 | **0.0952** |
| Cylinder | MSE | 4.44e-02 | 6.38e-03 | 9.31e-02 | 6.76e-02 | 4.13e-01 | **1.00e-04** |
| | RNE | 0.5976 | 0.7154 | 0.8644 | 0.7364 | 1.8201 | **0.0196** |
| Sediments | MSE | 3.61e-02 | 5.94e-03 | 7.10e-03 | 1.07e-01 | 2.00e-01 | **4.60e-04** |
| | RNE | 0.4759 | 0.6103 | 0.6673 | 0.8180 | 1.1186 | **0.0282** |
| Complex Obstacles | MSE | 5.33e-02 | 7.79e-03 | 6.09e-03 | 7.66e-02 | 2.97e-01 | **6.69e-05** |
| | RNE | 0.5061 | 0.6120 | 0.5410 | 0.6069 | 1.1956 | **0.0211** |
| Laser Heat | MSE | 2.48e+03 | 4.98e+03 | 3.88e+03 | 1.95e+03 | 6.78e+03 | **1.76e+01** |
| | RNE | 0.1208 | 0.1711 | 0.1510 | 0.1071 | 0.1998 | **0.0102** |

### 4.2.1 CLASSICAL PDEs

To validate the accuracy of DGNet as a general-purpose PDE solver, we first evaluate it on three well-established classical systems: the diffusion-dominated Allen–Cahn equation, the advection-driven Fisher–KPP equation, and the coupled FitzHugh–Nagumo system representing excitable media. These tasks span distinct physical regimes (diffusion, advection–reaction, and multi-variable coupling), and are widely used as canonical benchmarks in scientific machine learning.

Table 2 reports the quantitative results. DGNet consistently outperforms all baselines by a large margin across all equations, achieving up to one to two orders of magnitude lower mean squared error (MSE). This indicates that the proposed discrete Green formulation, combined with physics-informed operators, substantially reduces sample complexity and delivers markedly improved accuracy under limited supervision. Figure 3 provides qualitative comparisons. For the Allen–Cahn and Fisher–KPP equations, DGNet's predictions closely match the ground truth, whereas other models exhibit noticeable deviations, with MGN performing relatively better than the rest. The FitzHugh–Nagumo system is particularly challenging due to the formation of nonlinear spiral waves. Here, only DGNet successfully reproduces the propagation of spiral structures over long horizons, while baselines either dissipate the patterns or introduce severe distortions. These results show that even in a data-constrained setting, DGNet can accurately solve PDEs with fundamentally different underlying dynamics, highlighting its versatility and reliability as a data-efficient spatiotemporal solver.

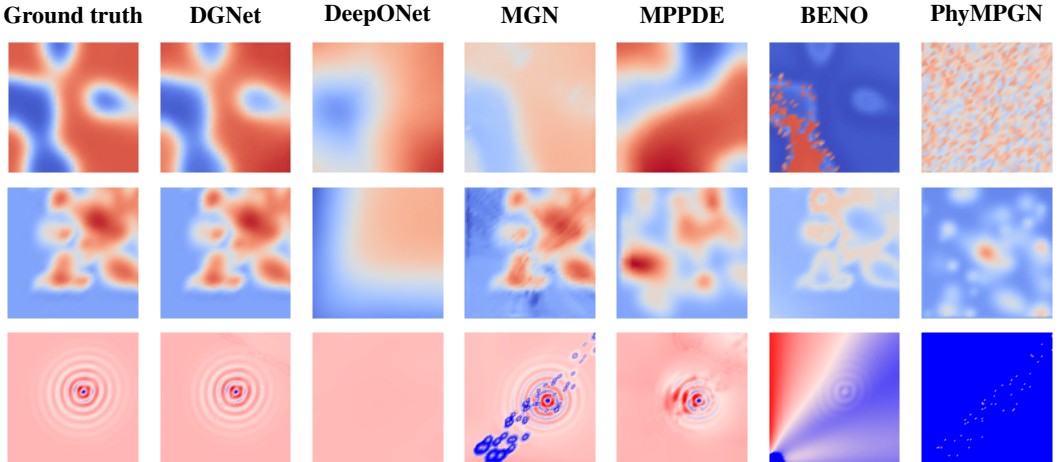

Figure 3: Visualization of prediction results on classical PDE scenarios. Rows from top to bottom correspond to the Allen–Cahn, FitzHugh–Nagumo, and Fisher–KPP equations, respectively.

### 4.2.2 COMPLEX GEOMETRIC DOMAINS

We next examine the ability of DGNet to handle PDEs defined on irregular meshes with complex boundaries. These scenarios simulate contaminant transport in a channel flow, with obstacles of varying shapes that induce rich flow structures. The three cases considered are: (i) a circular cylinder, (ii) sediment deposits on the walls, and (iii) a combination of elliptical obstacles and airfoil-shaped wall structures. Such settings are challenging due to intricate boundary conditions and highly nonuniform meshes.

Table 2 shows that DGNet achieves the lowest errors across all three domains, substantially outperforming baseline models. Figure 4 provides qualitative comparisons. In the cylinder scenario, DGNet accurately captures the Kármán vortex street in the wake of the obstacle, while other models exhibit distorted or dissipated vortical patterns. In the more complex sediment and obstacle cases, DGNet's predictions remain visually indistinguishable from ground truth, successfully reproducing contaminant filaments stretched and folded by the flow. By contrast, baseline methods either blur these fine-scale structures or fail to track their spatial location. These results demonstrate that DGNet, benefiting from the strong structural prior provided by the discrete Green formulation, achieves data-efficient learning and robust performance even on irregular geometries and nontrivial boundary conditions.

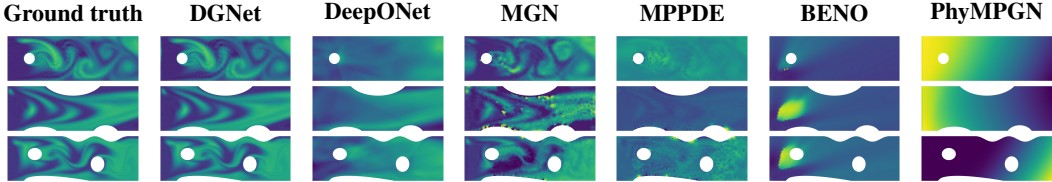

Figure 4: Visualization of prediction results on scenarios with complex geometric domains. Rows from top to bottom correspond to the cylinder, sediment, and complex obstacle cases, respectively.

### 4.2.3 GENERALIZATION TO UNSEEN SOURCE TERMS

A salient property of DGNet is its ability to generalize beyond source terms encountered during training. To test this, we consider a laser heat treatment task governed by the transient heat equation, where the source term $f(\mathbf{x}, t)$ corresponds to the spatiotemporally varying power density of moving laser beams. During training, the model is exposed to trajectories generated from a subset of source patterns. At test time, it must adapt to entirely novel laser paths, such as spline and Lissajous curves, that were never seen during training. This setup directly evaluates the zero-shot generalization capability to new forcing conditions, and serves as a stringent stress test of data efficiency under limited training coverage.

Quantitative results in Table 2 show that most baseline models suffer severe degradation when confronted with unseen source terms: their errors increase by several orders of magnitude compared to the training distribution. In contrast, DGNet maintains stable accuracy, with almost no performance drop relative to its results on seen source terms. Figure 5 further highlights this difference. DGNet accurately reproduces the spatiotemporal evolution of the temperature field, including the movement and spread of high-temperature regions, even under entirely novel laser trajectories. By comparison, baseline predictions collapse: the high-temperature regions become misplaced or overly diffused, failing to follow the true laser paths. These experiments confirm that DGNet uniquely addresses a critical limitation of existing neural PDE solvers, achieving robust generalization to unseen source terms—a scenario ubiquitous in real-world scientific and engineering applications.

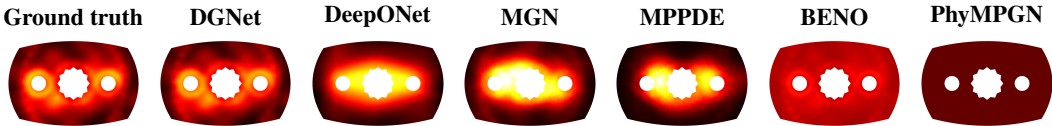

Figure 5: Visualization of generalization performance on the laser heat scenario with unseen sources.

### 4.3 ABLATION STUDY

We compare DGNet against four variants on the complex obstacle scenario: (A) **w/o $L_{physics}$**: removes the physics prior operator; (B) **w/o $L_{neural}$**: removes the neural correction operator; (C) **w/o Residual GNN**: removes the residual GNN while retaining the discrete Green solver; (D) **w/o Green**: replaces the discrete Green solver with a generic end-to-end GNN that only consumes operator features. The results are shown in Figure 6. Among all variants, **w/o Green** suffers the largest performance drop, highlighting the central role of the discrete Green solver as a strong structural prior and inductive bias in the model. Removing $L_{physics}$ degrades performance more severely than removing $L_{neural}$, demonstrating that the physics prior provides essential structural knowledge while the neural correction mainly fine-tunes discretization errors. Finally, the decline in the **w/o Residual GNN** variant confirms the residual GNN's effectiveness in capturing additional dynamics and improving overall accuracy.

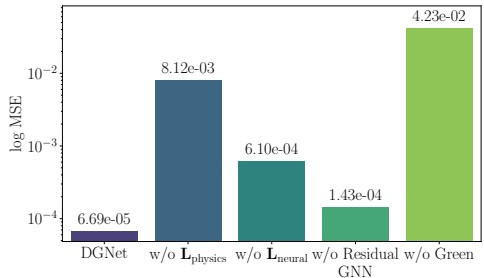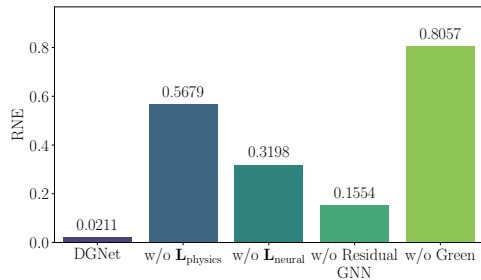

Figure 6: Ablation study on the complex obstacle scenario.

## 5 CONCLUSION

We introduced DGNet, a discrete Green network designed to improve the data efficiency of neural PDE solvers through structural inductive bias. By discretizing Green's function on graphs, DGNet transforms a classical continuous tool into a practical learning framework that preserves the superposition principle and reduces the burden of rediscovering solution operator structure from data. Coupled with physics-based operators and GNN corrections, this architecture provides a principled interface between physical fidelity and learnable flexibility. Extensive experiments across three categories of spatiotemporal PDE systems demonstrate that DGNet consistently achieves state-of-the-art accuracy under limited training trajectories, and maintains stable performance when generalizing to unseen source terms. Looking ahead, several open challenges remain, including extending the framework to quasilinear PDEs where the superposition principle no longer holds, and scaling to large three-dimensional systems. We view these directions as promising avenues for advancing discrete Green methods and broadening the impact in scientific machine learning.

ACKNOWLEDGMENT

Q.Yao is supported by Beijing Natural Science Foundation (under Grant No. 4242039). Y. Wang is sponsored by Beijing Nova Program.

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

## A    Derivation of the Discrete Green Solver

For completeness, we outline the derivation of the discrete Green solver and its relation to implicit time integration. Unlike explicit propagation matrices, which are dense and costly to construct, our approach realizes the Green operator implicitly through sparse linear solves.

### A.1    Explicit vs. Implicit Green Formulation

The discrete propagation matrix $\mathbf{G}(\Delta t)$ formally maps the state $\mathbf{u}^k$ to $\mathbf{u}^{k+1}$ and serves as the discrete analogue of the continuous Green's function:

$$\mathbf{u}^{k+1} = \mathbf{G}(\Delta t)\mathbf{u}^k.$$

Each column of $\mathbf{G}(\Delta t)$ corresponds to the system response from a unit impulse at one node, while each row represents how contributions from all nodes aggregate to update a given node. However, explicitly forming $\mathbf{G}(\Delta t)$ requires $O(N^2)$ storage and computation, which is infeasible for large meshes. We therefore derive an implicit realization based on stable time integration.

### A.2    Crank–Nicolson Derivation

Starting from the semi-discretized ODE system

$$\frac{d\mathbf{u}}{dt} = \mathbf{L}\mathbf{u} + \mathbf{f},$$

we apply the Crank–Nicolson scheme for stability and second-order accuracy. The time derivative is approximated by a central difference,

$$\frac{d\mathbf{u}}{dt} \approx \frac{\mathbf{u}^{k+1} - \mathbf{u}^k}{\Delta t},$$

while $\mathbf{L}\mathbf{u}$ and $\mathbf{f}$ are approximated by their averages at $t^k$ and $t^{k+1}$. This yields

$$\frac{\mathbf{u}^{k+1} - \mathbf{u}^k}{\Delta t} = \tfrac{1}{2}\big(\mathbf{L}\mathbf{u}^k + \mathbf{L}\mathbf{u}^{k+1}\big) + \tfrac{1}{2}\big(\mathbf{f}^k + \mathbf{f}^{k+1}\big).$$

Rearranging terms gives the sparse linear system

$$\left(\mathbf{I} - \tfrac{\Delta t}{2}\mathbf{L}\right)\mathbf{u}^{k+1} = \left(\mathbf{I} + \tfrac{\Delta t}{2}\mathbf{L}\right)\mathbf{u}^k + \tfrac{\Delta t}{2}\left(\mathbf{f}^k + \mathbf{f}^{k+1}\right),$$

which is equivalent to applying the discrete Green's function

$$\mathbf{G}(\Delta t) = \left(\mathbf{I} - \tfrac{\Delta t}{2}\mathbf{L}\right)^{-1}.$$

## A.3 EFFICIENT IMPLEMENTATION

In practice, we do not compute the inverse explicitly. Since $\mathbf{L}$ is sparse and depends only on the static mesh geometry, the coefficient matrix $\left(\mathbf{I} - \tfrac{\Delta t}{2}\mathbf{L}\right)$ is also sparse and remains fixed during rollout. We therefore adopt a "factorize once, solve many times" strategy: a single sparse LU factorization is performed before rollout, and forward/backward substitution reuses the factors at each time step. This reduces the per-step cost to nearly linear in the number of nonzeros, making the discrete Green solver scalable to large systems.

## B   DATASET DETAILS

We evaluate DGNet on three categories of PDE systems: (1) classical equations (Allen–Cahn, Fisher–KPP, FitzHugh–Nagumo), (2) contaminant transport in complex geometric domains, and (3) laser heat treatment with varying source terms. High-fidelity simulation data are generated with the FEniCSx finite element library on unstructured meshes and small time steps. Detailed parameters are reported in Table 3. Below we outline the dataset settings.

### B.1   CLASSICAL PDEs

**Allen–Cahn.**   Phase separation with $\epsilon = 0.04$ on $[0,1]^2$, $\Delta t = 0.005$, 1000 steps. Mesh: perturbed $35 \times 35$ grid ($\sim$1296 nodes). Initial condition: random noise in $[-0.5, 0.5]$. Boundary condition: homogeneous Neumann. Trajectories: 20 (10 train / 10 test).

**Fisher–KPP.**   Advection–reaction equation with $\rho = 1.0$, $\mathbf{c} = [0.1, 0.12]$, domain $[0,1]^2$. $\Delta t = 0.00125$, 1600 steps. Mesh: perturbed $40 \times 40$ grid ($\sim$1681 nodes). Initial condition: Gaussian blobs. Boundary: inflow Dirichlet, outflow natural. Trajectories: 20 (10 train / 10 test).

**FitzHugh–Nagumo.**   Two-component excitable system with parameters $D_u = 1/L_s^2$, $D_v = 6.181/L_s^2$, $\Gamma = 9.657$, $\alpha = 0.5$, $\beta = 2.1$, $L_s = 16\pi$. Domain $[0,1]^2$, $\Delta t = 0.005$, 1200 steps. Mesh: perturbed $99 \times 99$ grid ($\sim$10,000 nodes). Initial: circular perturbation. Boundary: inflow Dirichlet ($u = 0, v = 0$), others Neumann. Trajectories: 10 (7 train / 3 test).

### B.2   COMPLEX GEOMETRIC DOMAINS

All scenarios solve coupled Navier–Stokes and reaction–advection–diffusion equations. Fluid density $\rho = 1.0$, viscosity $\mu = 5 \times 10^{-3}$, contaminant $D = 2 \times 10^{-3}$, $\rho_g = 0.1$, $k_d = 0.025$. Simulation: $T = 100$s, $\Delta t = 0.05$s, 2000 steps. Trajectories: 20 (15 train / 5 test).

**Cylinder.**   Rectangular channel $[0, 24] \times [0, 8]$, obstacle: central circle ($r = 1.0$), $Re = 240$.

**Sediments.**   Same channel, three asymmetric smooth bumps attached to top/bottom walls.

**Complex Obstacles.**   Same channel, two ellipses + NACA airfoil-shaped bumps/grooves.

### B.3   LASER HEAT TREATMENT (VARYING SOURCE TERMS)

Transient heat conduction with $\rho = 7850$, $c_p = 450$, $k = 50$, $h_{\text{conv}} = 25$. Domain: complex plate with gear + circular holes. Boundary: convective, initial $T = 298.15$K. Heat source $Q(x,t)$: 10 moving lasers with random paths (orbit, spline, Lissajous, etc.). Simulation: $T = 60$s, $\Delta t = 0.5$s, 120 steps. Trajectories: 40 (20 train / 20 test).

Table 3: Simulation parameters of all datasets used in experiments.

| Dataset | Domain | Nodes (approx.) | $\Delta t$ | Steps | Traj. (train/test) |
|---------|--------|-----------------|-----------|-------|--------------------|
| *Classical PDEs* | | | | | |
| Allen–Cahn | $[0,1]^2$ | 1,296 | 0.005 | 1,000 | 10 / 10 |
| Fisher–KPP | $[0,1]^2$ | 1,681 | 0.00125 | 1,600 | 10 / 10 |
| FitzHugh–Nagumo | $[0,1]^2$ | 10,000 | 0.005 | 1,200 | 7 / 3 |
| *Complex Geometric Domains* | | | | | |
| Cylinder | $[0,24] \times [0,8]$ | 2,275 | 0.05 | 2,000 | 15 / 5 |
| Sediments | $[0,24] \times [0,8]$ | 5,758 | 0.05 | 2,000 | 15 / 5 |
| Complex Obstacles | $[0,24] \times [0,8]$ | 8,841 | 0.05 | 2,000 | 15 / 5 |
| *Varying Source Term* | | | | | |
| Laser Heat | Complex Plate | 6,072 | 0.5 | 120 | 20 / 20 |

## C  EXPERIMENTAL SETUP

All experiments are conducted on a workstation with four NVIDIA RTX 4090 (24GB) GPUs. Training typically completes within hours to one day depending on dataset size. Ground-truth PDE data are generated using FEniCSx (dolfinx 0.9.0). DGNet is implemented in PyTorch 2.5.1 with CUDA 12.4 and Python 3.13.5.

### C.1  TRAINING HYPERPARAMETERS

All models are trained with Adam and a step LR scheduler. We adopt trajectory segmentation for efficiency, and apply the loss only to the first and last predictions of each segment to balance single-step accuracy and long-term stability. Table 4 lists the hyperparameters for each experimental scenario (the three contaminant transport cases share identical settings).

Table 4: Training hyperparameters for each experimental scenario.

| Scenario | Batch Size | Initial LR | Epochs | Decay Step | Decay Rate | Sub-seq. Length | Train/Test Trajs. |
|----------|-----------|-----------|--------|-----------|-----------|-----------------|-------------------|
| Allen–Cahn | 36 | 1e-3 | 600 | 200 | 0.1 | 20 | 10/10 |
| Fisher–KPP | 24 | 5e-4 | 300 | 60 | 0.1 | 10 | 10/10 |
| FitzHugh–Nagumo | 10 | 1e-3 | 400 | 100 | 0.1 | 10 | 7/3 |
| Cylinder | 6 | 5e-4 | 300 | 100 | 0.2 | 6 | 15/5 |
| Sediments | 6 | 5e-4 | 300 | 100 | 0.2 | 6 | 15/5 |
| Complex Obstacles | 6 | 5e-4 | 300 | 100 | 0.2 | 6 | 15/5 |
| Laser Heat | 8 | 5e-4 | 400 | 200 | 0.1 | 8 | 20/20 |

### C.2  IMPLEMENTATION DETAILS

To solve the sparse linear system in Eq. 6, $\mathbf{A}\mathbf{u}_{\text{inter}}^{k+1} = \mathbf{b}$, we employ a factorize-once strategy with GPU-accelerated sparse algebra.

**Pre-computation.** The system matrix $\mathbf{A} = (\mathbf{I} - \frac{\Delta t}{2}\mathbf{L})$ depends only on mesh geometry and $\Delta t$, so it is built once before training. It is converted to a CuPy sparse matrix and factorized using `cupy.sparse.linalg.splu`, producing LU factors cached as `A_lu`.

**Forward pass.** At each step, the cached LU factors are used to efficiently solve $\mathbf{A}\mathbf{u}_{\text{inter}}^{k+1} = \mathbf{b}$ by forward/backward substitution, avoiding explicit matrix inversion and enabling fast long-term rollouts.

**Backward pass.** We implement a custom `torch.autograd.Function` with an adjoint formulation. Gradients are obtained by solving $\mathbf{A}^T \mathbf{g}_b = \mathbf{g}_{u,\text{inter}}$ using the cached LU factors, ensuring differentiability and efficiency.

# D MORE EXPERIMENTAL RESULTS

## D.1 VISUALIZATION OF LEARNED OPERATOR **L**

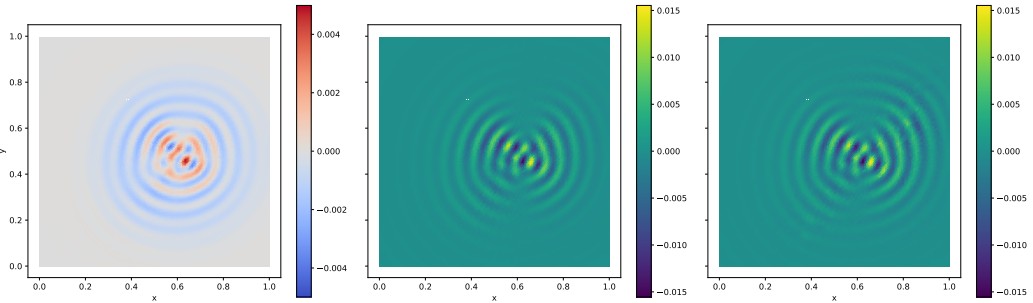

Figure 7: Comparison of the learned operator with the reference. Left: input physical field. Center: reference operator computed by finite difference. Right: operator predicted by DGNet.

Figure 7 compares the operator predicted by our model (right) with the reference computed by a finite difference scheme (center), given the same input physical field (left). The learned operator closely matches the reference in both structural patterns and numerical scale, demonstrating that DGNet accurately captures operator-level dynamics.

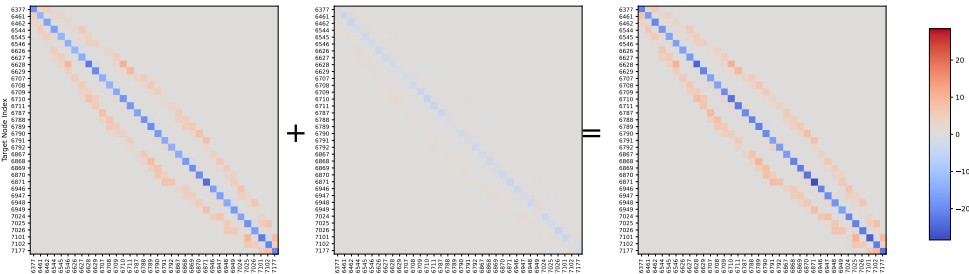

Figure 8: Heatmap visualization of operator matrices. Left: $\mathbf{L}_{\text{physics}}$. Middle: $\mathbf{L}_{\text{neural}}$. Right: final corrected operator **L**.

Figure 8 shows heatmaps of $\mathbf{L}_{\text{physics}}$ (left), $\mathbf{L}_{\text{neural}}$ (middle), and the final corrected operator **L** (right). The magnitude of entries in $\mathbf{L}_{\text{neural}}$ is significantly smaller than in $\mathbf{L}_{\text{physics}}$, indicating that the neural component mainly provides fine-grained corrections on top of the physical prior.

## D.2 INFERENCE COMPUTATIONAL OVERHEAD AND THROUGHPUT

We evaluate the computational efficiency of DGNet on a single NVIDIA RTX 4090 GPU. Table 5 details the computational costs across all datasets. Notably, once factorized, the per-step inference is highly efficient, achieving throughputs exceeding 90–130 steps/s across all scenarios.

Table 5: DGNet inference complexity and throughput measurements.

| Dataset | Nodes (N) | Pre-comp. Time (s) | Inference Time (ms/step) | Total Inf. Time (s) | Throughput (steps/s) | Peak Mem. (MB) |
|---|---|---|---|---|---|---|
| Allen–Cahn | 1,296 | 0.6274 | 7.7419 | 8.3693 | 129.17 | 152.60 |
| Fisher–KPP | 1,681 | 3.6413 | 7.5764 | 15.7636 | 131.99 | 227.97 |
| FitzHugh–Nagumo | 10,000 | 10.7362 | 10.8723 | 23.7829 | 91.98 | 3830.72 |
| Cylinder | 2,275 | 2.5194 | 7.7895 | 18.0984 | 128.38 | 368.81 |
| Sediments | 5,758 | 6.0459 | 7.7943 | 21.6345 | 128.30 | 1570.49 |
| Complex Obstacles | 8,841 | 9.6110 | 9.7367 | 29.0844 | 102.70 | 3258.10 |
| LaserHeat | 6,072 | 1.9382 | 7.7363 | 2.8666 | 129.26 | 1570.65 |

To assess scalability, Figure 9 illustrates the inference time and peak memory usage as the mesh size increases up to our largest experimental settings. The results demonstrate that DGNet maintains predictable and effective scaling behavior.

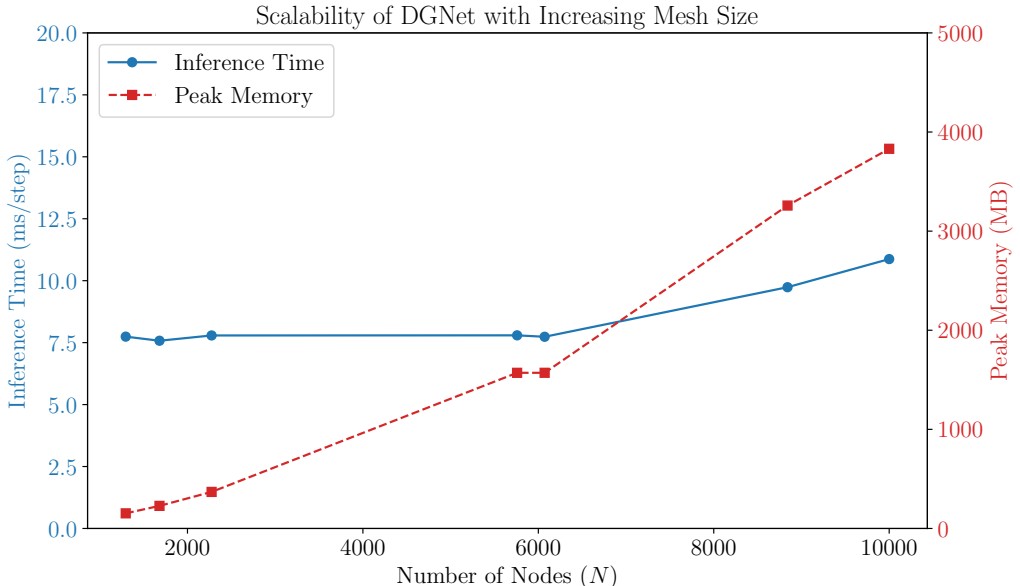

Figure 9: Scalability of inference time and peak memory with respect to mesh size.

Finally, Table 6 compares the single-trajectory inference time of DGNet against baseline models. Despite using pre-factorization, DGNet's inference time remains competitive with baseline methods. These results confirm that DGNet achieves superior accuracy without imposing a significant computational burden compared to existing methods.

Table 6: Comparison of single-trajectory inference time (in seconds) between DGNet and baseline methods.

| Dataset | DeepONet | MGN | MP-PDE | BENO | PhyMPGN | DGNet |
|---|---|---|---|---|---|---|
| Allen-Cahn | 1.33 | 5.12 | 5.96 | 14.06 | 6.71 | 8.37 |
| Fisher-KPP | 2.77 | 12.74 | 19.20 | 75.92 | 17.40 | 15.76 |
| FitzHugh-Nagumo | 106.52 | 15.48 | 45.23 | 67.32 | 10.72 | 23.78 |
| Cylinder | 6.38 | 20.15 | 24.90 | 133.01 | 12.47 | 18.10 |
| Sediments | 58.94 | 21.47 | 31.25 | 101.10 | 14.59 | 21.63 |
| Complex Obstacles | 141.92 | 23.59 | 47.36 | 108.96 | 20.31 | 29.08 |
| Laser Heat | 4.22 | 2.56 | 3.95 | 2.30 | 1.73 | 2.87 |

## D.3 EXTENDED ABLATION STUDIES

To verify the robustness of our architectural components across diverse physical regimes, we extend the ablation study to two additional scenarios: the FitzHugh-Nagumo system and the Laser Heat experiment. We compare DGNet against the same four variants defined in Section 4.3: w/o $L_{physics}$, w/o $L_{neural}$, w/o Residual GNN, and w/o Green GNN.

The results, presented in Figure 10 and Figure 11, are consistent with the findings in the main text. In both scenarios, the w/o Green GNN variant exhibits the most severe performance degradation, confirming that the discrete Green's function formulation is the fundamental backbone of our solver. Furthermore, removing the physics prior (w/o $L_{physics}$) leads to a substantial increase in error, highlighting the necessity of embedding physical laws directly into the operator. Finally, both the neural correction ($L_{neural}$) and the residual module contribute to the final accuracy, ensuring the model captures complex dynamics that pure discretization operators cannot resolve.

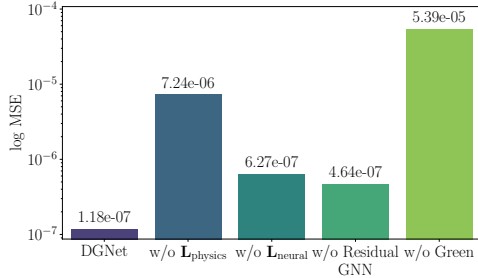 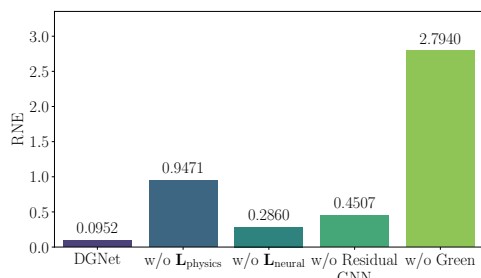

Figure 10: Ablation study results on the FitzHugh–Nagumo scenario.

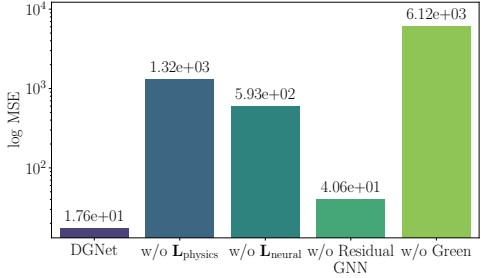 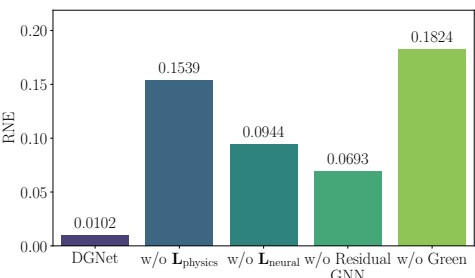

Figure 11: Ablation study results on the Laser Heat scenario.

### D.4 EMPIRICAL SPECTRUM PLOTS

To empirically corroborate the stability analysis presented in Appendix E, we visualize the eigenvalue spectra of the operators on the FitzHugh–Nagumo and Laser Heat datasets. In Figures 12 and 13, the blue markers represent the eigenvalues of the fixed physics prior $\mathbf{L}_{physics}$, while the orange markers represent the eigenvalues of the final learned hybrid operator $\mathbf{L} = \mathbf{L}_{physics} + \mathbf{L}_{neural}$.

The plots demonstrate that the eigenvalues of the total operator remain strictly confined to the left half of the complex plane ($\text{Re}(\lambda) < 0$), respecting the stability boundary indicated by the red dashed line. This confirms that the neural correction $\mathbf{L}_{neural}$ functions as a stable perturbation, preserving the dissipative structure of the physical system while refining its dynamics, in alignment with our theoretical stability guarantees.

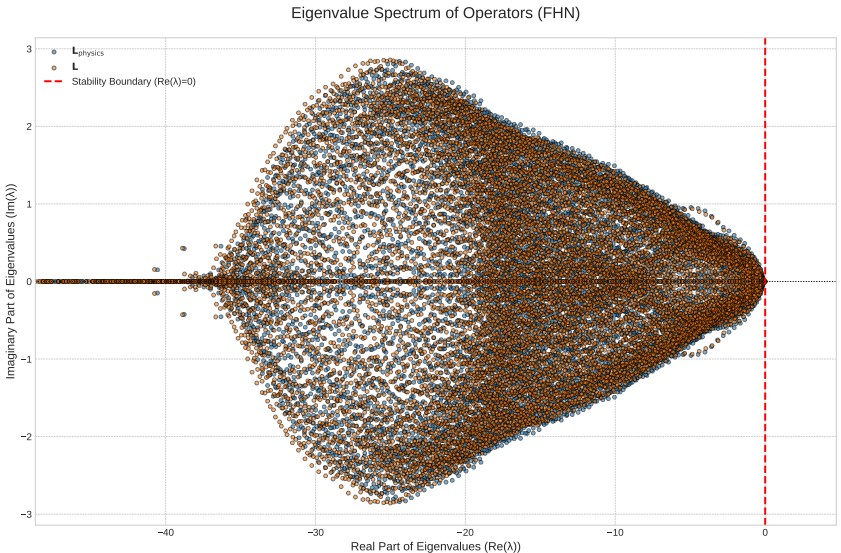

Figure 12: Eigenvalue spectrum analysis for the FitzHugh–Nagumo task. Blue indicates the physics prior $\mathbf{L}_{physics}$, and orange indicates the total hybrid operator $\mathbf{L}$.

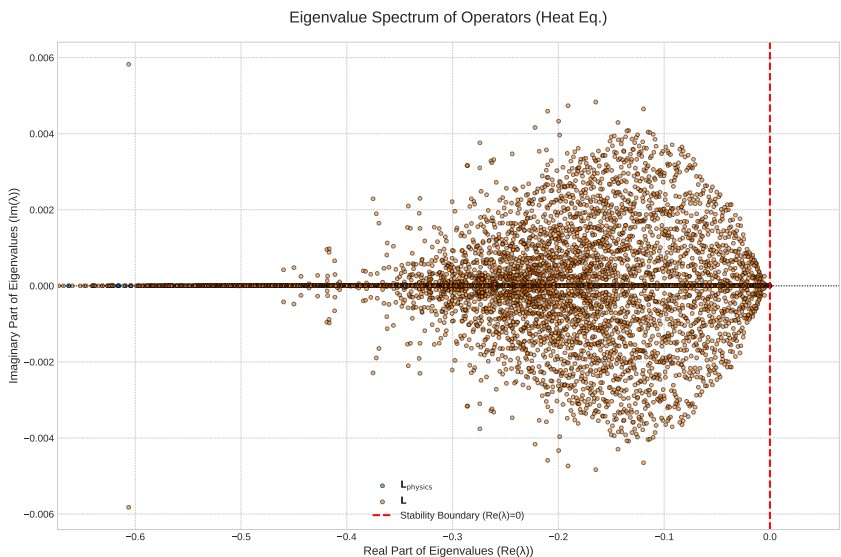

Figure 13: Eigenvalue spectrum analysis for the Laser Heat task. Blue indicates the physics prior $\mathbf{L}_{physics}$, and orange indicates the total hybrid operator $\mathbf{L}$.

### D.5 SPEED-ACCURACY TRADE-OFF ANALYSIS

To further contextualize the efficiency of our approach, we analyze the speed-accuracy trade-off on the FitzHugh-Nagumo dataset. We compare DGNet against a classical high-fidelity numerical solver, which employs the Finite Element Method (FEM) with a Newton nonlinear solver. By varying the mesh resolution, we establish a baseline curve linking computational cost to simulation error.

As shown in Figure 14, the classical numerical method exhibits a steep increase in inference time as the error tolerance decreases . In contrast, DGNet demonstrates a characteristic "flat" cost curve; its inference time is determined by the network architecture and remains constant regardless of the achieved precision. Crucially, in the high-precision regime (MSE $\approx 10^{-6}$), DGNet achieves comparable accuracy to the fine-mesh numerical solver but at a fraction of the computational cost. This highlights the distinct advantage of neural solvers: they do not aim to replace classical methods for infinite-precision tasks but serve as highly efficient surrogates within a practical accuracy range.

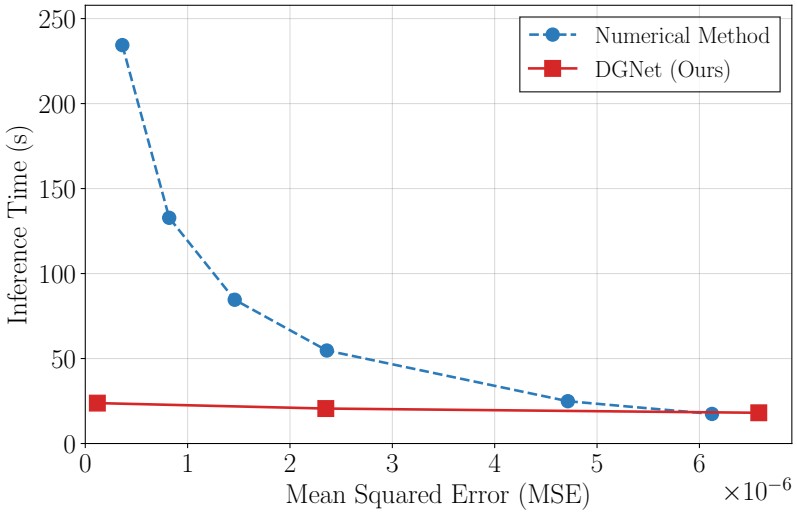

Figure 14: Speed-accuracy comparison between DGNet and the classical FEM numerical solver on the FitzHugh-Nagumo task.

## E    THEORETICAL ANALYSIS OF NEURAL CORRECTION OPERATOR $\mathbf{L}_{\text{NEURAL}}$

This section focuses on the theoretical analysis of the learned neural correction operator $\mathbf{L}_{\text{neural}}$. We highlight two core properties of this learnable component: Consistency and Stability. For the former, we show that the optimization process learns to cancel the local truncation error of the physical operator, thereby recovering the consistency of the discrete scheme. For the latter, we establish spectral constraints for $\mathbf{L}_{\text{neural}}$ based on the Numerical Range. As long as the correction term satisfies specific norm bounds, the hybrid operator maintains the A-stability of the physical system.

We first provide clear definitions and the discrete scheme, similar to Section 3: Let $\Omega \subset \mathbb{R}^d$ be a bounded domain, and $\mathcal{T}_h$ be an irregular mesh with characteristic size $h$. Our graph is constructed based on this mesh $\mathcal{T}_h$ (using the subscript $h$ to emphasize the characteristic scale). Discretizing the PDE $\partial_t u = \mathcal{L}u + f$ into a semi-discrete ODE system on a finite-dimensional space $\mathbb{R}^N$, the evolution of DGNet is governed by the hybrid operator $L_h \in \mathbb{R}^{N \times N}$:

$$L_h = L_{\text{physics}} + L_{\text{neural}}(\theta),$$

where $L_{\text{physics}}$ is the fixed discretization operator based on mesh geometry, and $L_{\text{neural}}(\theta)$ is the GNN correction term controlled by parameters $\theta$. The single-step inference of the system follows

the Crank–Nicolson implicit update scheme:

$$u^{k+1} = \left(I - \frac{\Delta t}{2}L_{\mathrm{h}}\right)^{-1}\left(I + \frac{\Delta t}{2}L_{\mathrm{h}}\right)u^k + \mathcal{S}(f),$$

where $G_h(\Delta t) = \left(I - \frac{\Delta t}{2}L_{\mathrm{h}}\right)^{-1}$ is the discrete Green's propagation operator, and $\mathcal{S}(f)$ is the source term response.

### E.1 CONSISTENCY ANALYSIS

Our consistency analysis aligns with the data-driven discretization framework proposed by Bar-Sinai et al. (2019). They demonstrated that learning correction coefficients on a coarse grid effectively captures the unresolved sub-grid physics (truncation errors), enabling high-fidelity simulation at low resolution.

**Definition 1 (Spatial Truncation Error).** Let $P_h$ be the projection operator from the continuous space to the discrete space. The local truncation error $\tau_h(u)$ of the physical operator $L_{\mathrm{physics}}$ with respect to the true operator $\mathcal{L}$ is defined as:

$$\tau_h(u) = L_{\mathrm{physics}}P_h u - P_h(\mathcal{L}u).$$

On irregular meshes, due to mesh skewness, there is typically a non-zero low-order consistency error, i.e., $\|\tau_h\| \not\to 0$.

**Theorem 1 (Consistency Guarantee).** For a neural solver with universal approximation properties, if the training objective is to minimize the single-step prediction error $\mathcal{J}(\theta) = \|u_{GT}^{k+1} - \hat{u}^{k+1}\|_2^2$ (where $\hat{u}$ denotes the model prediction and $u_{GT}$ denotes the ground truth), then the optimal neural correction term $L_{\mathrm{neural}}^*$ satisfies:

$$L_{\mathrm{neural}}^* P_h u \approx -\tau_h(u),$$

thereby eliminating the geometrically induced error of the physical discretization and recovering consistency.

*Proof.* Consider the single-step evolution from $t_k$ to $t_{k+1}$. The true solution $u_{GT}$ satisfies the semi-discrete equation with truncation error:

$$\frac{u_{GT}^{k+1} - u_{GT}^k}{\Delta t} = \frac{1}{2}L_{\mathrm{physics}}(u_{GT}^{k+1} + u_{GT}^k) + \frac{1}{2}\tau_h(u) + \dots$$

The predicted solution $\hat{u}^{k+1}$ of DGNet follows the update rule:

$$\frac{\hat{u}^{k+1} - u_{GT}^k}{\Delta t} = \frac{1}{2}(L_{\mathrm{physics}} + L_{\mathrm{neural}})(\hat{u}^{k+1} + u_{GT}^k) + \dots$$

Comparing the two equations, the consistency residual $E_{cons}$ of the hybrid operator relative to the true dynamics is:

$$\begin{aligned}
E_{cons} &= \|(L_{\mathrm{physics}} + L_{\mathrm{neural}})P_h u - P_h \mathcal{L}u\| \\
&= \|(L_{\mathrm{physics}}P_h u - P_h \mathcal{L}u) + L_{\mathrm{neural}}P_h u\| \\
&= \|\tau_h(u) + L_{\mathrm{neural}}P_h u\|.
\end{aligned}$$

During training, we minimize the prediction loss $\mathcal{J}(\theta)$, which is equivalent to finding a correction operator such that $L_{\mathrm{neural}}(\theta^*)P_h u \to -\tau_h(u)$. This implies that the neural network is explicitly driven to learn the negative of the truncation error, thereby canceling the geometric error and recovering the consistency of the numerical scheme. $\square$

### E.2 STABILITY ANALYSIS

Our stability analysis relies on the theory of Numerical Range. Following Trefethen & Embree (2005), we use the spectral enclosure property of the numerical range to bound the system's dynamics. To establish strict energy stability, we employ the Logarithmic Norm of the operator. As detailed in Wanner & Hairer (1996), a negative logarithmic norm $\mu[L_{\mathrm{h}}] < 0$ is a sufficient condition for unconditional stability in stiff systems. Our proof is based on this fact.

**Definition 2 (Numerical Range and Logarithmic Norm).** The numerical range of a matrix $A$ is defined as $W(A) = \{\frac{\langle x, Ax\rangle}{\langle x, x\rangle} \mid x \in \mathbb{C}^N \setminus \{0\}\}$, which possesses the following key properties:

- Spectral Enclosure: All eigenvalues $\lambda(A)$ of the matrix are contained within the numerical range, i.e., $\sigma(A) \subseteq W(A)$.

- Dissipativity Bound: The supremum of the real part of the numerical range equals the logarithmic norm: $\sup \operatorname{Re}(W(A)) = \mu_2(A) = \lambda_{\max}(\frac{A+A^T}{2})$.

**Theorem 2 (Stability Guarantee).** Assume the physical operator $L_{\text{physics}}$ is coercive (dissipative), i.e., for any $x \neq 0$, $\operatorname{Re}\langle x, L_{\text{physics}}x \rangle \leq -\eta \|x\|^2$ (where $\eta > 0$ is determined by the physical system). If the output of the neural correction operator $L_{\text{neural}}$ is constrained by architecture (e.g., bounded activation function tanh) such that its spectral norm satisfies $\|L_{\text{neural}}\|_2 \leq \gamma$, and the condition $\gamma < \eta$ holds, then the hybrid operator $L_{\text{h}} = L_{\text{physics}} + L_{\text{neural}}$ satisfies:

- Spectral Stability: The real parts of all eigenvalues are strictly negative, i.e., $\operatorname{Re}(\lambda(L_{\text{h}})) < 0$.

- Energy Stability: The discrete evolution operator $\left(I - \frac{\Delta t}{2}L_{\text{h}}\right)^{-1}\left(I + \frac{\Delta t}{2}L_{\text{h}}\right)$ is strictly contractive, i.e., $\|\left(I - \frac{\Delta t}{2}L_{\text{h}}\right)^{-1}\left(I + \frac{\Delta t}{2}L_{\text{h}}\right)\|_2 < 1$.

*Proof.* For any unit vector $x$ ($\|x\|_2 = 1$), the quadratic form corresponding to the hybrid operator is:
$$\langle x, L_{\text{h}}x \rangle = \langle x, L_{\text{physics}}x \rangle + \langle x, L_{\text{neural}}x \rangle.$$
Taking the real part and using the Cauchy-Schwarz inequality:
$$\operatorname{Re}\langle x, L_{\text{h}}x \rangle = \operatorname{Re}\langle x, L_{\text{physics}}x \rangle + \operatorname{Re}\langle x, L_{\text{neural}}x \rangle.$$
Due to the coercive dissipativity of $L_{\text{physics}}$ and the boundedness of $L_{\text{neural}}$:
$$\operatorname{Re}\langle x, L_{\text{h}}x \rangle \leq -\eta + |\langle x, L_{\text{neural}}x \rangle| \leq -\eta + \|L_{\text{neural}}\|_2 \leq -\eta + \gamma.$$

This indicates that the numerical range $W(L_{\text{h}})$ lies entirely to the left of the line $x = -\eta + \gamma$ in the complex plane. Therefore, by the spectral enclosure property $\sigma(L_{\text{h}}) \subseteq W(L_{\text{h}})$, we directly obtain the upper bound for the real parts of all eigenvalues:
$$\max_{\lambda \in \sigma(L_{\text{h}})} \operatorname{Re}(\lambda) \leq \sup \operatorname{Re}(W(L_{\text{h}})) \leq -\eta + \gamma.$$

This proves the spectral stability of the operator $L_{\text{h}}$.

For energy stability, note that the norm of the Crank-Nicolson propagation operator $\left(I - \frac{\Delta t}{2}L_{\text{h}}\right)^{-1}\left(I + \frac{\Delta t}{2}L_{\text{h}}\right)$ satisfies the following bound:
$$\|\left(I - \frac{\Delta t}{2}L_{\text{h}}\right)^{-1}\left(I + \frac{\Delta t}{2}L_{\text{h}}\right)\|_2 \leq \frac{1 + \frac{\Delta t}{2}\mu_2(L_{\text{h}})}{1 - \frac{\Delta t}{2}\mu_2(L_{\text{h}})}.$$

Since $\mu_2(L_{\text{h}}) = \sup \operatorname{Re}(W(L_{\text{h}})) \leq -\eta + \gamma < 0$, we know that the above norm is less than 1, obtaining energy stability.

Therefore, as long as the magnitude $\gamma$ of the neural correction term is smaller than the physical dissipation rate $\eta$, the hybrid operator of DGNet possesses a strictly left-half-plane spectral distribution and unconditional A-stability. $\square$

# F SUPPLEMENTARY EXPERIMENT

## F.1 EXPERIMENTS WITH 3D SCENES

To evaluate DGNet's scalability to high-dimensional geometries and complex boundary conditions, we constructed a 3D transient heat conduction benchmark.

**Geometry and Physical Model.** The domain is a mechanical bracket (L-shaped support with a central bore) discretized into an unstructured tetrahedral mesh with approximately 6,000 nodes (see Figure 15). We solve the transient heat equation with convective cooling:

$$\rho c_p \partial_t T = \nabla \cdot (k \nabla T) - h_{\text{conv}}(T - T_{\text{amb}}) + Q(\mathbf{x}, t),$$

using the following parameters: $\rho = 7850 \, \text{kg} \, \text{m}^{-3}$, $c_p = 450 \, \text{J} \, \text{kg}^{-1} \text{K}^{-1}$, $k = 50 \, \text{W} \, \text{m}^{-1} \text{K}^{-1}$, and $h_{\text{conv}} = 25 \, \text{W} \, \text{m}^{-2} \text{K}^{-1}$. The simulation spans $T = 60$,s with a time step $\Delta t = 0.5$,s.

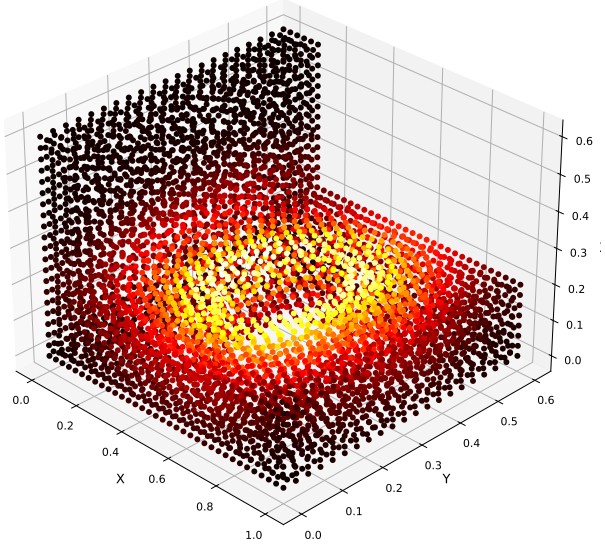

Figure 15: Visualization of the 3D heat conduction scenario, showing the mechanical bracket geometry.

**Laser Source Model.** The source term $Q(\mathbf{x}, t)$ models a superposition of 10 moving Gaussian laser spots. To rigorously test generalization, we generate trajectories that combine deterministic paths (mimicking industrial processing) and stochastic paths (random splines and reflections). The beam radius $\sigma$ is randomized in $[0.5, 2.5]$,mm, and power profiles are temporally ramped to create multi-modal forcing.

**Dataset and Results.** We generated 60 trajectories in total, split into 20 for training and 40 for testing. The experimental results are shown in Table 7, DGNet significantly outperforms baseline methods. This confirms that our framework generalizes effectively to complex 3D geometries.

Table 7: Quantitative results on the 3D laser-heat task. DGNet achieves the lowest MSE and RNE compared to baselines.

| Metric | DeepONet | MGN | MP-PDE | BENO | PhyMPGN | DGNet |
|--------|----------|-----|--------|------|---------|-------|
| MSE | 4.41e+03 | 4.60e+02 | 2.59e+04 | 5.28e+02 | 8.83e+03 | 4.42e+01 |
| RNE | 0.1630 | 0.1249 | 0.1993 | 0.0938 | 0.2352 | 0.0314 |

F.2    EXPERIMENTS WITH COMPLEX GEOMETRY AND HIGHER REYNOLDS NUMBERS

To further test the robustness of the model, we constructed a more challenging variant of the Contaminant Transport scenarios presented in Section 4.2.2 Experimental Setup. We maintained the identical governing equations, source injection strategy, and time-step settings as the previous experiments. The modifications were strictly limited to two aspects:

- **Complex Boundaries:** The channel walls were modified from straight to sinusoidal (wavy) forms to force continuous flow redirection.
- **Higher Reynolds Number:** The Reynolds number was increased to $Re = 300$, pushing the flow further into the transitional regime with richer vortex shedding dynamics.

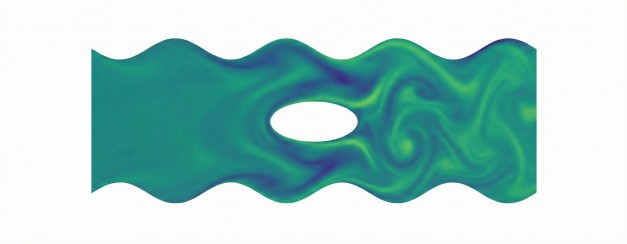

Figure 16: Visualization of the modified wavy channel scenario ($Re = 300$).

**Dataset and Results.** We generated 20 trajectories under this configuration, and the scene visualization is shown in Figure 16. Experimental results are shown in Table 8. Despite the more complex geometry, DGNet maintained robust performance.

Table 8: Quantitative comparison on the high-Reynolds ($Re = 300$) fluid and pollutant transport task.

| Metric | DeepONet | MGN | MP-PDE | BENO | PhyMPGN | DGNet |
|--------|----------|-----|--------|------|---------|-------|
| MSE | 2.38e-01 | 3.59e-03 | 7.94e-03 | 2.30e-02 | 7.95e-02 | **6.29e-04** |
| RNE | 0.8539 | 0.3591 | 0.4729 | 0.6292 | 1.1838 | **0.0385** |

F.3    EXPERIMENTS WITH LARGE-SCALE LIGHT-DRIVEN REACTIONS

To validate the model's scalability and generalization capability, we introduce a Light-Driven Chemical Reaction benchmark.

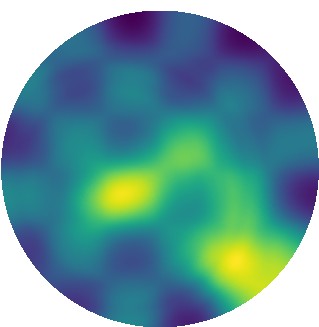

Figure 17: Snapshot of the light-driven reaction scenario.

**Simulation Setup.** The domain is a circular Petri dish ($R = 1$) discretized into a fine unstructured mesh containing approximately 30,000 nodes. We solve a transient reaction-diffusion equation with zero-flux boundary conditions:

$$\frac{\partial C}{\partial t} = D\nabla^2 C - kC + f(\mathbf{x}, t),$$

where $D = 5 \times 10^{-4}$ and the decay rate $k = 5 \times 10^{-3}$.

**Complex Dynamic Source.** The source term $f(\mathbf{x}, t)$ represents dynamic light intensity driving the reaction. To prevent overfitting to simple distributions, we construct the source by superimposing six distinct dynamic patterns (including rotating spirals, pulsating rings, moving spots, and traveling waves) with randomized parameters. This results in a highly nonlinear and unpredictable spatiotemporal field.

**Dataset and Results.** We significantly expanded the dataset size to 200 trajectories (40 for training, 160 for testing), spanning $T = 100$s with $\Delta t = 0.5$s. The experimental results are shown in Table 9, DGNet demonstrates the best accuracy and scalability. It can scale to large systems and generalize to unseen complex source terms.

Table 9: Quantitative results on the large-scale light-driven reaction task.

| Metric | DeepONet | MGN | MP-PDE | BENO | PhyMPGN | DGNet |
|--------|----------|-----|--------|------|---------|-------|
| MSE | 6.11e+03 | 1.93e+02 | 1.37e+01 | 6.04e+02 | 2.87e+02 | 3.97e+00 |
| RNE | 0.8832 | 0.1582 | 0.0422 | 0.2796 | 0.1921 | 0.0227 |

### F.4 MORE SOURCE TERM GENERALIZATION EXPERIMENTS

To evaluate DGNet's robustness to variations in source characteristics in a zero-shot setting (the trained model is fixed), we performed robustness tests that vary three key source attributes: spatial bandwidth (spot size), amplitude (peak power), and the number of concurrent sources. Each attribute was scaled relative to the training distribution by factors $\{0.5, 1, 2, 3\}$ and evaluation was performed on the Laser-Heat 3D scenario.

Table 10 summarizes representative numeric results (MSE) under these perturbations. DGNet generalizes effectively across the tested range. These findings indicate strong robustness to changes in source characteristics.

Table 10: Robustness to source characteristics. Each column scales the corresponding attribute by the indicated factor relative to training.

| Source Attribute | $0.5\times$ | $1\times$ | $2\times$ | $3\times$ |
|------------------|-------------|-----------|-----------|-----------|
| bandwidth | 1.53e+01 | 1.76e+01 | 3.88e+01 | 7.36e+01 |
| amplitude | 2.27e+01 | 1.76e+01 | 1.62e+01 | 1.95e+01 |
| number of concurrent sources | 1.09e+01 | 1.76e+01 | 3.26e+01 | 4.40e+01 |

### F.5 FAILURE MODE: QUASILINEAR PHASE TRANSITION

To probe regimes where the linear superposition assumption may break, we constructed a controlled failure-mode experiment based on a quasilinear phase-transition heat equation:

$$\rho c_{\text{eff}}(T)\partial_t T = \nabla \cdot \big(k(T)\nabla T\big) + Q(x, t), \tag{11}$$

where both the effective heat capacity $c_{\text{eff}}(T)$ and thermal conductivity $k(T)$ undergo sharp, temperature-dependent changes (e.g., step changes modeling phase transitions or abrupt material property shifts). This introduces strong nonlinear coupling between temperature and transport that violates linear superposition.

We employed temperature-dependent parametrizations with a sharp transition centered near a critical temperature $T_c$. Figure 18 presents a visual comparison between the ground truth and the DGNet prediction. As observed in the Ground Truth (left), the temperature field exhibits a distinct plateau (uniform red region). This accurately reflects the physical mechanism of phase transition, where the effective heat capacity $c_{eff}(T)$ surges at $T_c$, causing the system to absorb input energy as latent heat rather than increasing the local temperature. In contrast, the DGNet Prediction (right) fails to capture this nonlinear saturation effect. Instead, it predicts a diffusive distribution with high-temperature hotspots (bright yellow regions). This behavior occurs because the architecture relies on the superposition of source responses; it treats the energy injection as additive linear heating,

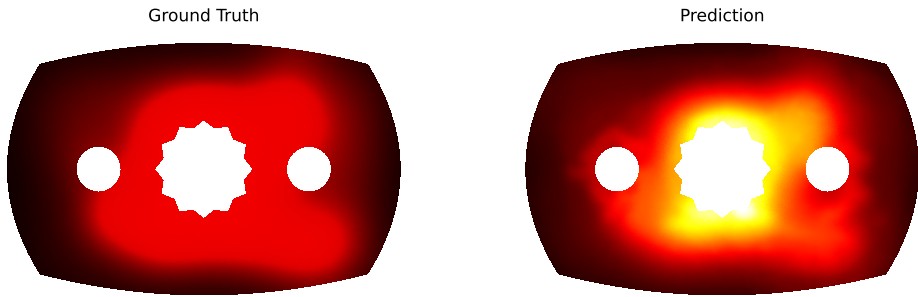

Figure 18: Visualization of the failure mode in the quasilinear Phase Transition scenario.

ignoring the drastic, state-dependent changes in material properties. Consequently, prediction errors concentrate heavily around the phase transition interface and regions of source overlap.

These results empirically confirm the boundary of our method: while DGNet is highly effective for semilinear systems, the linear superposition assumption limits its applicability in quasilinear regimes characterized by strong state-dependent parameter coupling.

### F.6 COMPARISON WITH RECENT BASELINES

To comprehensively position DGNet against recent advancements, we evaluated three baselines SINGER (Feng et al., 2025), AMG-GNN (Li et al., 2025), and Transolver (Wu et al., 2024a), which represent multi-scale architecture, graph-based architecture and Transformer architecture, respectively.

The results summarized in Table 11 show that DGNet achieves substantially lower error rates across all evaluated metrics, while recent baselines exhibit noticeably degraded performance under the limited-data setting considered in this work. As discussed earlier, we attribute this performance gap primarily to data efficiency. Although these data-driven models are powerful and expressive, they typically rely on thousands of training trajectories to accurately capture complex dynamics. In contrast, by embedding physical priors through the discrete Green's function, DGNet is able to generalize effectively from limited data, whereas purely data-driven baselines lack sufficient inductive bias.

Table 11: Quantitative comparison with recent state-of-the-art baselines (SINGER, AMG-GNN, Transolver) on representative tasks.

| Dataset | Metric | SINGER | AMG (Harnessing) | Transolver | DGNet |
|---|---|---|---|---|---|
| FitzHugh-Nagumo | MSE | 4.58e-06 | 3.30e-05 | 7.87e-06 | **1.18e-07** |
|  | RNE | 1.3844 | 2.4837 | 0.6729 | **0.0952** |
| Complex Obstacles | MSE | 9.41e-03 | 1.40e-03 | 8.94e-04 | **6.69e-05** |
|  | RNE | 0.7746 | 0.3058 | 0.1855 | **0.0211** |
| Laser Heat | MSE | 9.58e+03 | 2.96e+03 | 5.72e+03 | **1.76e+01** |
|  | RNE | 0.2504 | 0.1831 | 0.1785 | **0.0102** |

