# OpenReview forum: "DGNet: Discrete Green Networks for Data-Efficient Learning of Spatiotemporal PDEs"
_ICLR.cc/2026/Conference — ICLR 2026 Poster_

### Official Review · Reviewer_pbbn · 2025-10-27

**Soundness:** 2
**Presentation:** 2
**Contribution:** 2
**Rating:** 4
**Confidence:** 4

**Summary:**

The paper proposes **DGNet (Discrete Green Network)** for learning **spatiotemporal PDEs** on irregular meshes, with a core design that **explicitly decouples system evolution from source response** via a **discrete Green’s function** formulation. Concretely, it discretizes ( $\partial_t u = L_x[u] + f$ ) into a **Crank–Nicolson** update and realizes a per-step Green operator ($G(\Delta t) = (I-\tfrac{\Delta t}{2}L)^{-1}$), enabling superposition of (i) state propagation and (ii) source-term response. The spatial operator is built as a **physics–neural hybrid** ($L=L_{\text{physics}}+L_{\text{neural}}$): ($L_{\text{physics}}$) uses mesh-aware gradient/Laplacian discretizations, while ($L_{\text{neural}}$) is a GNN correction for discretization errors; a lightweight residual GNN further captures leftover dynamics. An efficient **factorize-once, solve-many** sparse LU scheme makes long rollouts practical. Experiments across **classical PDEs**, **complex geometries**, and **unseen source terms** show consistent SOTA accuracy, with marked robustness when test-time sources differ from training.

**Strengths:**

1. **Addresses challenging spatiotemporal PDEs on irregular meshes.**
   The setting combines *both* complex geometry (unstructured meshes) and temporal evolution, which is closer to real CFD/physical simulations than static or grid-regular cases. Tackling this regime is practically meaningful.

2. **Green’s-function–motivated decomposition.**
   By formulating updates through a discrete Green operator, the method **explicitly separates** (i) the evolution of the initial state and (ii) the accumulated response to time-varying sources. This superposition-friendly design improves interpretability, allows cleaner handling of unseen source patterns, and provides a principled bridge between physics and learning.

3. **Physics-informed design on irregular meshes.**
   The hybrid operator and loss terms embed **physically consistent priors** (e.g., mesh-aware differential operators, stability-friendly time-stepping), encouraging fidelity to the governing equations while letting the learned components correct discretization errors. This tends to enhance robustness, boundary handling, and generalization across meshing/sampling changes.

**Weaknesses:**

1. **Heavy reliance on pre-factorization; fairness not quantified.**
   The method hinges on a **“factorize once, solve many”** sparse LU of ($(I-\tfrac{\Delta t}{2}L)$) before rollout, then reuses the factors each step. While efficient per step, this adds a non-negligible **precomputation and memory** burden uncommon in NN-only baselines, yet there’s **no cost table** (params/FLOPs/GPU memory/wall-clock) to establish fairness. The paper mentions hardware (4×RTX4090) and vague training durations (“hours to one day”) but lacks matched-budget reporting across methods. A standardized efficiency comparison is needed.

2. **Small problem sizes vs. stated motivation.**
   Core 2D meshes are relatively modest (e.g., **Cylinder ~2.3k nodes**, Sediments ~5.8k, Complex Obstacles ~8.8k; classical PDEs 1.3k–10k). These scales undercut the claim of handling challenging irregular domains; results on **larger meshes/3D** or higher Reynolds/complex boundaries would better substantiate practical robustness.

3. **Baseline scope/positioning leave doubts.**
   The baseline set (DeepONet, MGN, MP-PDE, PhyMPGN, **BENO**) mixes operator, graph, and hybrid methods, but (i) it’s unclear **how comparability is enforced** beyond a sentence (“tuned to have comparable parameters/training costs”) and (ii) some choices may be **mismatched to spatiotemporal settings** (e.g., BENO targets elliptic PDEs), while **strong recent multiscale/graph/transformer/point-operator** [1-5] baselines on irregular domains are missing. A head-to-head with closer, stronger contemporaries and a **capacity-controlled** comparison would sharpen novelty and fairness.

[1] Feng, Mingquan, et al. "SINGER: Stochastic Network Graph Evolving Operator for High Dimensional PDEs." The Thirteenth International Conference on Learning Representations. 2025.

[2] Zhang, Xuan, et al. "SineNet: Learning Temporal Dynamics in Time-Dependent Partial Differential Equations." The Twelfth International Conference on Learning Representations.

[3] Wang, Qi, et al. "P $^ 2$ C $^ 2$ Net: PDE-Preserved Coarse Correction Network for efficient prediction of spatiotemporal dynamics." Advances in Neural Information Processing Systems 37 (2024): 68897-68925.

[4] Li, Zhihao, et al. "Harnessing scale and physics: A multi-graph neural operator framework for pdes on arbitrary geometries." Proceedings of the 31st ACM SIGKDD Conference on Knowledge Discovery and Data Mining V. 1. 2025.

[5] Wu, Haixu, et al. "Transolver: A Fast Transformer Solver for PDEs on General Geometries." International Conference on Machine Learning. PMLR, 2024.

**Questions:**

1. **Precomputation & fairness.**
   Please report a cost table (params, FLOPs/step, GPU memory, LU pre-factorization time for Eq. (9), wall-clock/epoch, hardware) and confirm matched training budgets across baselines.

2. **Scalability & factorization reuse.**
   Results on substantially larger meshes (and ideally 3D) would align with the motivation. Also clarify when (L) changes across samples (coefficients/BCs/mesh): must you re-factor each time, and what is the end-to-end impact?

3. **Benchmarks & baseline selection.**
   Add stronger, recent multiscale/transformer/graph operator baselines for irregular spatiotemporal PDEs, and justify inclusion of elliptic-focused baselines. For **cylinder flow**, reconcile why baseline errors are far worse than commonly reported (detail resolution/Re/splits/targets/configs).

4. **Robustness & hybrid ablations.**
   Provide stress tests (larger $(\Delta t)$, stiff sources, mesh noise/density shifts) and ablations separating $(L_{\text{physics}})$ vs. $(L_{\text{neural}})$ to show stability and the benefit of the physics–neural hybrid.

**Details Of Ethics Concerns:**

No ethics concerns.

---

> ### Author Response · Authors · 2025-11-25
>
> We sincerely thank the reviewer for the thoughtful and detailed feedback.
> We have carefully considered every point raised and revised the manuscript accordingly.
> Below we provide point-by-point responses, and all corresponding changes in the updated manuscript
> are highlighted in blue. We genuinely hope that our clarifications address the reviewer’s concerns,
> and we would be happy to provide further information if needed.
>
> > **W1 & Q1:** Heavy reliance on pre-factorization. Pre-factorization adds a non-negligible precomputation and memory burden.
> >
> > Fairness not quantified. Please report a cost table and confirm matched training budgets across baselines.
>
> **A1:**
>
> **Pre-computation Overhead:** Regarding the specific overhead of pre-decomposition, please refer to the table below, which lists the computational overhead and throughput (measured on a single RTX 4090 graphics card). This table is included in Appendix D.3 of the updated version. It can be seen that even on the largest datasets, pre-computation takes only a few seconds.
>
> | Dataset           | Nodes ($N$) | Pre-comp. Time (s) | Inference Time (ms/step) | Total Inference Time (s) | Throughput (steps/s) | Peak Memory (MB) |
> | ----------------- | ----------- | ------------------ | ------------------------ | ------------------------ | -------------------- | ---------------- |
> | Allen-Cahn        | 1,296       | 0.62               | 7.74                     | 8.36                     | 129.17               | 152.60           |
> | Fisher-KPP        | 1,681       | 3.64               | 7.57                     | 15.76                    | 131.99               | 227.97           |
> | FitzHugh-Nagumo   | 10,000      | 10.73              | 10.87                    | 23.78                    | 91.98                | 3830.72          |
> | Cylinder          | 2,275       | 2.51               | 7.78                     | 18.09                    | 128.38               | 368.81           |
> | Sediments         | 5,758       | 6.04               | 7.79                     | 21.63                    | 128.30               | 1570.49          |
> | Complex Obstacles | 8,841       | 9.61               | 9.73                     | 29.08                    | 102.70               | 3258.10          |
> | Laser Heat        | 6,072       | 1.93               | 7.73                     | 2.86                     | 129.26               | 1570.65          |
>
> Furthermore, the table below provides the single-trajectory inference time (in seconds) for DGNet and all baseline models on different datasets. This table has been included in Appendix D.3. As shown, DGNet exhibits inference time on the same scale as other baselines. Pre-factorization does not lead to a significant decrease of inference efficiency.
>
> |                   | DeepONet | MGN   | MP-PDE | BENO   | PhyMPGN | DGNet |
> | ----------------- | -------- | ----- | ------ | ------ | ------- | ----- |
> | Allen-Cahn        | 1.33     | 5.12  | 5.96   | 14.06  | 6.71    | 8.37  |
> | Fisher-KPP        | 2.77     | 12.74 | 19.20  | 75.92  | 17.40   | 15.76 |
> | FitzHugh-Nagumo   | 106.52   | 15.48 | 45.23  | 67.32  | 10.72   | 23.78 |
> | Cylinder          | 6.38     | 20.15 | 24.90  | 133.01 | 12.47   | 18.10 |
> | Sediments         | 58.94    | 21.47 | 31.25  | 101.10 | 14.59   | 21.63 |
> | Complex Obstacles | 141.92   | 23.59 | 47.36  | 108.96 | 20.31   | 29.08 |
> | Laser Heat        | 4.22     | 2.56  | 3.95   | 2.30   | 1.73    | 2.87  |

---

> ### Author Response · Authors · 2025-11-25
>
> (Cont.)
>
> **Training cost & Fairness:** To comprehensively demonstrate training costs and prove the fairness of the baseline comparison, we list the detailed training costs of DGNet on different datasets (Hardware: 4 $\times$ RTX 4090).
>
> | Dataset           | Nodes ($N$) | Params (M) | FLOPs/step (G) | Pre-comp. Time (s) | Wall-clock/epoch (s) | Memory Util  (GB) |
> | ----------------- | ----------- | ---------- | -------------- | ------------------ | -------------------- | ----------------- |
> | Allen-Cahn        | 1,296       | 0.35       | 0.09    | 0.62               | 79.13        | 22.8     |
> | Fisher-KPP        | 1,681       | 0.35       | 0.13    | 3.54               | 101.02       | 23.4     |
> | FitzHugh-Nagumo   | 10,000      | 0.65       | 1.16    | 10.67              | 365.56       | 92.6      |
> | Cylinder          | 2,275       | 0.82       | 0.19    | 2.84               | 170.99       | 92.8     |
> | Sediments         | 5,758       | 0.82       | 0.64     | 6.21               | 247.31     | 90.6     |
> | Complex Obstacles | 8,841       | 0.82       | 1.21     | 9.47               | 382.75     | 88.2     |
> | LaserHeat         | 6,072       | 0.26       | 0.69     | 2.00               | 165.85     | 43.1      |
>
> We controlled the sample size and subsequence length to bring the total memory consumption close to the hardware limit (i.e., maximizing the utilization of the RTX 4090s) to achieve the fastest training. Please note that this does not represent the minimum memory required by the model.
>
> **Matching Budgets:** Furthermore, we list the parameter count comparison between DGNet and baseline models on three representative datasets.
>
> |                   | DGNet | DeepONet | MGN  | MPPDE | BENO | PhyMPGN |
> | ----------------- | ----- | -------- | ---- | ----- | ---- | ------- |
> | FitzHugh-Nagumo   | 652k  | 670k     | 646k | 702k  | 746k | 682k    |
> | Complex Obstacles | 827k  | 859k     | 909k | 814k  | 935k | 950k    |
> | LaserHeat         | 263k  | 295k     | 267k | 253k  | 254k | 282k    |
>
> We strictly controlled the experimental settings: all baseline models were tuned to have the same parameter scale and comparable memory usage as DGNet, and consistent training time was ensured. This ensures that performance differences stem from the method itself, rather than computational resource bias.
>
> > **W2 & Q2.1:** Small problem sizes. Core 2D meshes are relatively modest. Results on larger meshes/3D or higher Reynolds/complex boundaries would better substantiate practical robustness.
>
> **A2:**
> **larger Meshes/3D/Complex Boundaries Experiments:** For large-scale experiments, we add a new light-driven chemical reaction experimental scenario. This scenario employs a richer combination of test trajectories (approximately 200) and is expanded to a larger scale (nearly 30,000 nodes). The experimental results are shown below, and detailed settings can be found in Appendix F.3.
>
> | Metric | DeepONet | MGN      | MP-PDE   | BENO     | PhyMPGN  | DGNet    |
> | ------ | -------- | -------- | -------- | -------- | -------- | -------- |
> | MSE    | 6.11e+03 | 1.93e+02 | 1.37e+01 | 6.04e+02 | 2.87e+02 | **3.97e+00** |
> | RNE    | 0.8832     | 0.1582   | 0.0422   | 0.2796   | 0.1921   | **0.0227** |
>
> For 3D experiments, we have conducted transient heat conduction experiments in a 3D domain (containing approximately 6,000 nodes). As shown in the table below, DGNet maintains superior prediction accuracy in 3D geometries. For detailed experimental settings, please refer to Appendix F.1.
>
> | Metric | DeepONet | MGN      | MP-PDE   | BENO     | PhyMPGN  | DGNet        |
> | ------ | -------- | -------- | -------- | -------- | -------- | ------------ |
> | MSE    | 4.41e+03 | 4.60e+02 | 2.59e+04 | 5.28e+02 | 8.83e+03 | **4.42e+01** |
> | RNE    | 0.1630   | 0.1249   | 0.1993   | 0.0938   | 0.2352   | **0.0314**   |
>
> For complex boundaries experiments, we added a new CDR scenario test with a higher Reynolds number and more complex boundary conditions. The results are shown below, and details are in Appendix F.2.
>
> | Metric | DeepONet | MGN      | MP-PDE   | BENO     | PhyMPGN  | DGNet        |
> | ------ | -------- | -------- | -------- | -------- | -------- | ------------ |
> | MSE    | 2.38e-01 | 3.59e-03 | 7.94e-03 | 2.30e-02 | 7.95e-02 | **6.29e-04** |
> | RNE    | 0.8539   | 0.3591   | 0.4729   | 0.6292   | 1.1838   | **0.0385**   |
>
> **The Concern About Small Scale:** We need to clarify that compared to the works listed in W3, except for the ShapeNet Car scenario (32,186 nodes), their node counts in core irregular mesh benchmarks are typically only 1k to 10k (same scale as DGNet). As for the ShapeNet Car scenario with significantly higher nodes, it is a static or time-averaged problem that does not involve the time evolution of physical systems. Its complexity is far lower than systems with time evolution; thus, larger node scales in that context do not prove model scalability.

---

> ### Author Response · Authors · 2025-11-25
>
> > **Q2.2:** Factorization reuse. Clarify when $L$ changes across samples, must you re-factor each time, and what is the end-to-end impact?
>
> **A3:**
>
> **Factorization Reuse:** The variation of operator $L$ and its LU decomposition results depends only on mesh geometry and physical parameters/boundary types, and is completely independent of the source term $f(x,t)$ and initial condition $u(x,0)$.
>
> **End-to-End Impact:** Even if re-decomposition is needed in new scenarios, the few extra seconds added during pre-decomposition still bring an order-of-magnitude speedup compared to traditional numerical solvers. Even for neural network solvers, pre-processing time is not a bottleneck.
>
> > **W3 & Q3:** Baseline scope/positioning leave doubts. Some choices may be mismatched to spatiotemporal settings (e.g., BENO targets elliptic PDEs), while strong recent multiscale/graph/transformer/point-operator baselines on irregular domains are missing. For cylinder flow, reconcile why baseline errors are far worse than commonly reported.
>
> **A4:**
>
> **Comparison with Recent Baselines:** We have cited all suggested works in the updated paper. SineNet and P²C²Net are essentially CNN-based architectures unable to directly handle the irregular meshes focused on in this paper, so they are not suitable as direct baselines. For the remaining baselines, we compared them as follows (details in Appendix F.6):
>
> | Dataset           | Metric | SINGER   | AMG (Harnessing) | Transolver | DGNet        |
> | ----------------- | ------ | -------- | ---------------- | ---------- | ------------ |
> | FitzHugh-Nagumo   | MSE    | 4.58e-06 | 3.30e-05         | 7.87e-06   | **1.18e-07** |
> |                   | RNE    | 1.3844   | 2.4837           | 0.6729     | **0.0952**   |
> | Complex Obstacles | MSE    | 9.41e-03 | 1.40e-03         | 8.94e-04   | **6.69e-05** |
> |                   | RNE    | 0.7746   | 0.3058           | 0.1855     | **0.0211**   |
> | LaserHeat         | MSE    | 9.58e+03 | 2.96e+03         | 5.72e+03   | **1.76e+01** |
> |                   | RNE    | 0.2504   | 0.1831           | 0.1785     | **0.0102**   |
>
> While these baselines are powerful, they typically rely on training data containing thousands of trajectories to achieve optimal performance. However, our work introduces physical priors through Green's functions, enabling superior performance with fewer training trajectories (at most 20), whereas data-driven methods often degrade due to insufficient inductive bias.
>
> **Reason for Baseline Selection:** We considered a combination of baselines (DeepONet, MGN, PhyMPGN, etc.) aiming to cover mainstream directions like "Operator Learning," "Graph Networks," and "Physics-Informed" methods. This criterion is also reflected in the papers you listed. For example, P²C²Net only compared four baselines, three of which were classic FNO, UNet, and DeepONet, newer (2023) baseline models were only compared to PeRCNN. PhyMPGN also used a similar comparison method, which is common in this field.
>
> **Reason for Including BENO:** Although BENO was originally designed for elliptic PDEs, we included it because it is one of the few methods based on Green's function theory. Results show it is not entirely inapplicable; for example, in the FitzHugh-Nagumo scenario, BENO was the only baseline able to capture some wavefront features, demonstrating the potential of the Green's function idea.

---

> ### Author Response · Authors · 2025-11-25
>
> > **Q4:** Robustness & hybrid ablations. Provide stress tests and ablations separating $L_{physics}$ vs. $L_{neural}$ to show stability and the benefit of the physics–neural hybrid.
>
> **A5:**
>
> **Robustness:** To evaluate robustness, we conducted extensive zero-shot stress tests by varying source bandwidth, amplitude, and the number of concurrent sources from $0.5\times$ to $3\times$ the levels seen during training. As shown in the table below, DGNet maintains stable accuracy across these unseen conditions, demonstrating strong robustness to changes in source behavior. Detailed settings are provided in Appendix F.4.
>
> |                              | 0.5 ×    | 1 ×      | 2 ×      | 3 ×      |
> | ---------------------------- | -------- | -------- | -------- | -------- |
> | bandwidth                    | 1.53e+01 | 1.76e+01 | 3.88e+01 | 7.36e+01 |
> | amplitude                    | 2.27e+01 | 1.76e+01 | 1.62e+01 | 1.95e+01 |
> | number of concurrent sources | 1.09e+01 | 1.76e+01 | 3.26e+01 | 4.40e+01 |
>
> **Stress Tests:** The nonlinear reaction term in the FitzHugh–Nagumo system is well known to induce stiffness, and the Laser Heat scenario similarly involves stiff source behavior due to the rapid transients generated by the moving laser. Regarding mesh noise, both the Allen–Cahn and Fisher–KPP datasets were generated with explicit random mesh perturbations, directly demonstrating DGNet’s robustness to grid irregularities.
>
> **Ablation Experiments:** The original paper has already provided separate ablation results for $L_{physics}$ and $L_{neural}$. In addition, we add ablation studies under more scenarios, as shown in the table below. These results clearly demonstrate the incremental benefit yielded by the Residual GNN. Corresponding histograms have also been included in the appendix D.4.
>
> | Dataset         | Metric | DGNet    | w/o $L_{physics}$ | w/o $L_{neural}$ | w/o Residual GNN | w/o Green |
> | --------------- | ------ | -------- | ----------------- | ---------------- | ---------------- | --------- |
> | Laser Heat      | MSE    | 1.76e+01 | 1.32e+03          | 5.93e+02         | 4.06e+01         | 6.12e+03  |
> |                 | RNE    | 0.0102   | 0.1539            | 0.0944           | 0.0693           | 0.1824    |
> | FitzHugh-Nagumo | MSE    | 1.18e-07 | 7.24e-06          | 6.27e-07         | 4.64e-07         | 5.39e-05  |
> |                 | RNE    | 0.0102   | 0.1539            | 0.0944           | 0.0693           | 0.1824    |

---

> ### Author Response · Authors · 2025-11-27
>
> Dear Reviewer,
>
> Thank you again for the time you have spent evaluating our paper. As the deadline for the author–reviewer discussion is approaching, we would greatly appreciate any feedback you may be able to provide.
>
> We hope that our responses have addressed your concerns, and we would be glad to clarify anything further if needed.
>
> Thank you once again for your time and consideration.
>
> Best regards,
>
> The Authors

---

### Official Review · Reviewer_dKtN · 2025-10-31

**Soundness:** 3
**Presentation:** 2
**Contribution:** 3
**Rating:** 6
**Confidence:** 4

**Summary:**

The work proposes DGNet, a GNN-based model, to solve a class of PDEs. The method provides an explicit way to decouple the system evolution term from the effect of the source term based on the superposition principle of the PDE theory. The authors, in addition, investigate the relation between the Green’s function and a discrete time integration method, yielding the discrete counterpart of the superposition using the Green’s function. Further, the spatial derivative operator is expressed as a mixture of the classical numerical method and GNN. The experimental results show that the proposed method demonstrates the best accuracy among the considered baselines and generalization to unseen source terms. The authors conduct an ablation study to confirm that the proposed ingredients work properly.

**Strengths:**

1. The focus on the source term is interesting. Since the source/forcing term is used across a wide range of research and engineering applications, the work is of great importance to the community.
2. The method is simple and reasonable. The authors successfully leverage classical theory to develop an effective approach for elegantly handling the source term.

**Weaknesses:**

1. The applicable domain and limitations of the work are unclear. Since the method relies on the superposition and linear approximation of the spatial operator $\mathcal{L}_\boldsymbol{x}$, the reviewer considers it to work only for semilinear PDEs, although the paper says “PDE” with no specifications. The authors should clarify the assumptions and limitations of the work.
2. Section 3.4.1 is unclearly written. The paper uses $A_i$ and $d_i$ for volume and area, respectively, but this is outside the standard notation. In addition, there is no explanation about $N(i)$, $\alpha_{ij}$, and $\beta_{ij}$. The reviewer recommends adding a figure showing the variable settings for improved readability.
3. The experimental evaluation is weak. The work uses machine learning methods as baselines, but the comparison of computational speed and accuracy with a classical numerical solver should also be provided. In particular, since the method includes the scheme from the classical solver, it is not obvious that the proposed approach is more efficient than its classical counterpart. In addition, since machine learning methods have some error, the evaluation of a classical solver should be performed across multiple settings, e.g., varying the mesh resolution and convergence threshold, to capture the speed-accuracy tradeoff.

Minor point:

* Labels in Figure 5 and description in the main text are not aligned (e.g., w/o fit in the table vs w/o Residual GNN in the text). The reviewer recommends aligning them in the same word.

**Questions:**

1. The method assumes $u$ represents a scalar field (line 125), but can we extend it to vector and tensor fields?
2. How is the source term varied in the datasets? Can the method generalize to a source with a norm larger than that of the training dataset (e.g., a two-times larger source)? If not, the authors should explicitly present the limitations and the cases where the proposed model fails.
3. The work assumes the mesh is generated using the Delaunay triangulation. Why did the authors choose that configuration? Can we consider using other types of meshes, e.g., quadrilateral and polygonal meshes? How about 3D meshes? What would be the difficulty in extending the method?
4. The paper argues for the necessity of the neural correction operator for coarse or skewed meshes (line 265), but since the mesh is generated using the Delaunay triangulation, the reviewer considers there to be no skewed mesh in the present work. In addition, in the experiments, in particular, the FitzHugh–Nagumo dataset, the mesh resolution seems high enough. In that case, how does the author explain the necessity of the neural correction?

---

> ### Author Response · Authors · 2025-11-25
>
> We sincerely thank the reviewer for the constructive comments and the recognition of our work!
> Below we provide detailed point-by-point responses.
> For convenience, all corresponding revisions in the updated manuscript are highlighted in blue.
> Please let us know if any additional clarification is needed.
>
> > **W1:** Since the method relies on the superposition and linear approximation of the spatial operator $\mathcal{L}_{\mathbf{x}}$, the reviewer considers it to work only for semilinear PDEs.
>
> **A1:** We fully agree that the Discrete Green's Network architecture is primarily designed for semilinear PDEs. We have clearly defined this scope and assumptions in the Preliminary section, and we have identified quasilinear PDEs as an important direction for future research.
>
> > **W2:** Section 3.4.1 is unclearly written. The paper uses $A_i$ and $d_i$ for volume and area, respectively, but this is outside the standard notation. In addition, there is no explanation about $\mathcal{N}(i)$, $\alpha_{ij}$, and $\beta_{ij}$. The reviewer recommends adding a figure showing the variable settings for improved readability.
>
> **A2:** We appreciate the reviewer's professional suggestions. In the updated manuscript, we have standardized the notation for the control volume as $|\Omega_i|$. Additionally, we have explicitly defined the relative angles $\alpha_{ij}, \beta_{ij}$ and the neighborhood $\mathcal{N}(i)$, and added a figure showing the variable settings in the main text to improve readability.
>
> > **W3:** It is not obvious that the proposed approach is more efficient than its classical counterpart. The evaluation of a classical solver should be performed across multiple settings, to capture the speed-accuracy tradeoff.
>
> **A3:**
>
> **Efficiency of DGNet:** First, we clarify that DGNet has a significant speed advantage over traditional numerical solvers. The most time-consuming part—sparse LU decomposition—is performed only once before inference. Subsequent steps only require extremely fast calculation using cached factors and neural network forward passes. In contrast, traditional solvers (e.g., FEM) typically require time-consuming linear system solving or even nonlinear iterations at every time step.
>
> To demonstrate DGNet's practicality, we report the computational overhead and throughput in the table below (measured on a single RTX 4090). The table has been included in Appendix D.3 in the updated manuscript. It demonstrates DGNet's inference efficiency. For example, on the Complex Obstacles dataset, to reach an MSE error of $10^{-5}$ level, numerical solver needs thousands of nodes and takes nearly one hour. DGNet takes only 30 seconds, showing a massive speedup at this precision level.
>
> | Dataset           | Nodes ($N$) | Pre-comp. Time (s) | Inference Time (ms/step) | Total Inference Time (s) | Throughput (steps/s) | Peak Memory (MB) |
> | ----------------- | ----------- | ------------------ | ------------------------ | ------------------------ | -------------------- | ---------------- |
> | Allen-Cahn        | 1,296       | 0.62         | 7.74           | 8.36                     | 129.17               | 152.60           |
> | Fisher-KPP        | 1,681       | 3.64         | 7.57           | 15.76                    | 131.99               | 227.97           |
> | FitzHugh-Nagumo   | 10,000      | 10.73        | 10.87          | 23.78                    | 91.98                | 3830.72          |
> | Cylinder          | 2,275       | 2.51         | 7.78           | 18.09                    | 128.38               | 368.81           |
> | Sediments         | 5,758       | 6.04         | 7.79           | 21.63                    | 128.30               | 1570.49          |
> | Complex Obstacles | 8,841       | 9.61         | 9.73           | 29.08                    | 102.70               | 3258.10          |
> | Laser Heat        | 6,072       | 1.93          | 7.73          | 2.86                     | 129.26               | 1570.65          |
>
> **Speed-Accuracy Trade-off:** To further address concerns, we plotted a trade-off curve on the FitzHugh-Nagumo dataset by adjusting mesh resolution, please see Appendix D.6. In this scenario, numerical methods show a linear convergence cost, but DGNet is still significantly faster at higher precision. On the Complex Obstacles dataset, the difference in efficiency is so huge that it is difficult to even draw a comparable trade-off curve.
>
> **Design Philosophy:** This phenomenon reflects the design philosophy of Neural PDE solvers. Unlike numerical methods, which can significantly increase accuracy by increasing cost, neural solvers have a flatter inference time curve that does not change drastically with precision. However, they can significantly accelerate PDE solving within an acceptable accuracy range. **Neural PDE solvers do not replace traditional solvers but serve as a complement, with different design philosophies and applicable scenarios.**

---

> ### Author Response · Authors · 2025-11-25
>
> > **W4:** Labels in Figure 5 and description in the main text are not aligned (e.g., w/o fit in the table vs w/o Residual GNN in the text). The reviewer recommends aligning them in the same word.
>
> **A4:** We have corrected this inconsistency. We updated the labels in the Figure to "w/o Residual GNN" to align with the description in the main text.
>
> > **Q1:** The method assumes $u$  represents a scalar field (line 125), but can we extend it to vector and tensor fields?
>
> **A5:** Yes, the DGNet architecture is naturally extensible to both vector and tensor fields.
>
> **Vector Fields:** Taking a two-dimensional vector field $(u, v)$ as an example, Eq. 4 becomes a system of ODEs containing two state components. The spatial operator $L$ expands into a $2N \times 2N$ block matrix to explicitly capture cross-component coupling. The core Crank-Nicolson discretization logic remains invariant, yielding the similar update formula (Eq. 7) applied to the concatenated state vector $[u, v]$. While this scales memory usage with the number of components, the GNN adapters simply adjust by increasing the input feature dimensions.
>
> **Tensor Fields:** Tensor fields (e.g., stress tensors) are computationally treated as multi-channel vector fields. By flattening all independent tensor components into a unified state vector and encoding their interactions within the extended operator $L$, the solution algorithm remains mathematically consistent without modification.
>
> > **Q2:** How is the source term varied in the datasets? Can the method generalize to a source with a norm larger than that of the training dataset?
>
> **A6:** In the original training dataset, the variations in source terms are mainly reflected in the spatial diversity of movement paths, while the intensity remains relatively constant.
>
> Regarding whether the model can generalize to source terms with norms larger than the training set, the answer is yes. We conducted extensive zero-shot stress tests by varying source bandwidth, amplitude, and the number of concurrent sources from $0.5\times$ to $3\times$ the levels seen during training. As shown in the table below, DGNet maintains stable accuracy across these unseen conditions, demonstrating strong robustness to changes in source behavior. Detailed settings are provided in Appendix F.4.
>
> |                              | 0.5 ×    | 1 ×      | 2 ×      | 3 ×      |
> | ---------------------------- | -------- | -------- | -------- | -------- |
> | bandwidth                    | 1.53e+01 | 1.76e+01 | 3.88e+01 | 7.36e+01 |
> | amplitude                    | 2.27e+01 | 1.76e+01 | 1.62e+01 | 1.95e+01 |
> | number of concurrent sources | 1.09e+01 | 1.76e+01 | 3.26e+01 | 4.40e+01 |

---

> ### Author Response · Authors · 2025-11-25
>
> > **Q3:** The work assumes the mesh is generated using the Delaunay triangulation. Why did the authors choose that configuration? Can we consider using other types of meshes, e.g., quadrilateral and polygonal meshes? How about 3D meshes? What would be the difficulty in extending the method?
>
> **A7:**
>
> **Delaunay Triangulation:** We adopt Delaunay triangulation because it is one of the most mature and widely used standards for generating unstructured meshes. Its key property of maximizing the minimum angle effectively avoids poorly shaped elements and improves numerical stability. Moreover, many discrete differential operators on triangular meshes (e.g., the cotangent Laplacian) have well-established theoretical convergence guarantees, making them particularly suitable as reliable physical priors in our framework.
>
> **Extension to Quadrilateral/Polygonal Meshes:** DGNet's graph neural network architecture is topology-agnostic, thus fully supporting quadrilateral or polygonal meshes. The only technical change lies in the construction of $L_{physics}$: we need to replace the cotangent weight formula with a discrete format applicable to arbitrary polygons. This theory is quite mature in computational geometry. Although the construction process is slightly more complex, it provides high-accuracy physical priors with no theoretical obstacles.
>
> **Extending to 3D Meshes:** We have conducted transient heat conduction experiments in a 3D domain (containing approximately 6,000 nodes). As shown in the table below, DGNet maintains superior prediction accuracy in 3D geometries. For detailed experimental settings, please refer to Appendix F.1.
>
> | Metric | DeepONet | MGN      | MP-PDE   | BENO     | PhyMPGN  | DGNet        |
> | ------ | -------- | -------- | -------- | -------- | -------- | ------------ |
> | MSE    | 4.41e+03 | 4.60e+02 | 2.59e+04 | 5.28e+02 | 8.83e+03 | **4.42e+01** |
> | RNE    | 0.1630   | 0.1249   | 0.1993   | 0.0938   | 0.2352   | **0.0314**   |
>
> The core challenge in extending to large-scale 3D problems lies in computational resource requirements rather than methodology. Specifically, although the matrix $L$ remains sparse in 3D, the size of the separator in the nested dissection in sparse direct decomposition increases from $O(N^{0.5})$ in 2D to $O(N^{0.67})$ in 3D. According to sparse matrix theory, this causes the time complexity of pre-decomposition to surge from $O(N^{1.5})$ to $O(N^2)$. More critically, the memory footprint of the fill-in increases from $O(N \log N)$ to $O(N^{1.33})$, rendering the method unscalable. Therefore, in our future work, we plan to introduce iterative solvers such as Imcomplete LU (ILU) or AMG to replace the current direct solver, in order to adapt to large-scale 3D simulations.
>
> > **Q4:** The reviewer considers there to be no skewed mesh in the present work. In the experiments, in particular, the FitzHugh–Nagumo dataset, the mesh resolution seems high enough. How does the author explain the necessity of the neural correction?
>
> **A8:** We thank the reviewer for their meticulous review and insightful question. We answer the question point by point below.
>
> - **Skewed Meshes:** We fully agree with the point that Delaunay triangulation does indeed guarantee the geometric quality of the mesh, eliminating severely skewed meshes.
> - **Necessity of $L_{neural}$:** We believe the necessity of $L_{neural}$ should be understood from the perspective of discretization approximation. Real physical operators act on continuous space, while $L_{physics}$ approximates them using discrete operators. Although a high-quality mesh reduces approximation errors, an objective gap always exists between discrete and continuous operators. The core role of $L_{neural}$ is to capture and correct this residual, thereby improving the approximation accuracy of the operator without increasing mesh density. Our ablation experiments also confirm this: removing $L_{neural}$ directly leads to an increase in prediction error. Based on this, we have revised the wording in the paper.
> - **Correction in FitzHugh-Nagumo Dataset:** Although 10k nodes seem numerous, this resolution is not high relative to the drastically changing wavefront propagation dynamics. We can explain this from the perspective of characteristic scales. Based on experimental parameters, the ratio of Feature Scale / Mesh Spacing is approximately 3. Therefore, in numerical computation, this mesh density is not considered fine, making the neural correction particularly important.

---

> ### Author Response · Authors · 2025-11-27
>
> Dear Reviewer,
>
> Thank you again for the time you have spent evaluating our paper. As the deadline for the author–reviewer discussion is approaching, we would greatly appreciate any feedback you may be able to provide.
>
> We hope that our responses have addressed your concerns, and we would be glad to clarify anything further if needed.
>
> Thank you once again for your time and consideration.
>
> Best regards,
>
> The Authors

---

### Official Review · Reviewer_HxRT · 2025-11-01

**Soundness:** 3
**Presentation:** 3
**Contribution:** 3
**Rating:** 4
**Confidence:** 4

**Summary:**

This paper proposes DGNet, a discrete Green network framework for solving spatiotemporal PDEs with a particular focus on explicit modeling of source terms.
Unlike conventional neural PDE solvers that implicitly mix the source and system state, DGNet explicitly decouples the system evolution from the source response, inspired by the Green’s function formalism.
The model constructs a graph-based discrete operator that preserves the superposition principle. The operator $\mathcal{L}$ is composed of a physics-based component $\mathcal{L}{\text{physics}}$ and a neural correction component $\mathcal{L}{\text{NN}}$, where the former computes spatial derivatives such as gradients and Laplacians numerically, and the latter leverages a message passing neural network (MPNN) to correct approximation errors.
DGNet further incorporates Crank–Nicolson time integration to ensure stability in temporal evolution. Across multiple PDE benchmarks, including irregular mesh scenarios and novel source terms, DGNet achieves state-of-the-art accuracy and demonstrates strong robustness compared to existing operator-learning methods.

**Strengths:**

The paper addresses a longstanding limitation in neural PDE solvers—their inability to generalize to unseen source terms—by explicitly incorporating the Green’s function concept into a learnable, graph-based framework.
This formulation is conceptually elegant: by separating the effect of the source term from the system dynamics, DGNet captures the response structure in a principled and interpretable manner.
The combination of physics-based discretization and neural correction strikes a strong balance between numerical fidelity and data-driven flexibility. In particular, the hybrid operator $\mathcal{L} = \mathcal{L}{\text{physics}} + \mathcal{L}{\text{NN}}$ effectively merges computational stability with adaptability, while maintaining computational efficiency.
Empirical results show exceptionally high performance—often outperforming existing baselines by large margins—and stability on unseen forcing terms.
Overall, DGNet’s design provides a conceptually grounded and practically effective approach for learning PDE dynamics on irregular meshes and under novel source conditions.

**Weaknesses:**

Despite its strong results, the paper suffers from several clarity and interpretability issues that limit its accessibility.
First, the presentation of the operator $\mathcal{L}$ and its components lacks precision. Although equations define $\mathcal{L}{\text{physics}}$ and $\mathcal{L}{\text{NN}}$, it remains unclear how $\mathcal{L}{\text{physics}}$ is numerically computed and integrated into the overall update rule—whether gradients and Laplacians are used merely as input features or as direct numerical operators.
Similarly, the interaction between $\mathcal{L}{\text{NN}}$ (the MPNN correction) and the source term $f$ is only superficially discussed, leaving readers uncertain about how the model actually combines physical and learned dynamics to advance the PDE state $u$.
Although the abstract emphasizes source-term generalization, the main text provides limited explanation of how the method ensures this capability beyond architectural intuition.
Second, the neural component is relatively minimal. Apart from the residual MPNN correction, the rest of the solver heavily relies on standard numerical discretizations. While this hybrid structure is defensible, the role of the NN component could be elaborated to justify the “learning” aspect of the framework.
Third, the evaluation protocol raises concerns about robustness. Reported test sets are extremely small (3–20 samples), making it difficult to confidently assess generalization claims. The performance gains may partially stem from limited sampling rather than true model generalization.
Finally, after the numerical solution step, the exposition becomes sparse, with several mathematical derivations (Eqs. 5–7) presented without intuitive interpretation or discussion.

**Questions:**

- In the ablation study, does “w/o $\mathcal{L}_{\text{NN}}$” correspond to solving the PDE purely via numerical methods (i.e., without any learned correction)?
- How exactly is $\mathcal{L}_{\text{physics}}$ computed in practice—are the gradients and Laplacians incorporated as feature inputs to the MPNN, or directly as numerical differential operators?
- The paper emphasizes strong generalization to unseen source terms, but given the small test sizes (e.g., 3–20), can the authors provide additional experiments or statistical evidence to support this claim?
- How does DGNet compare to PHYMPGN (ICLR 2025) under identical experimental settings? The differences in reported performance seem unexpectedly large.
- Could larger-scale experiments (more trajectories or longer temporal horizons) be conducted to further validate generalization?

---

> ### Author Response · Authors · 2025-11-25
>
> We sincerely thank the reviewer for the positive assessment and the detailed feedback.
> We truly appreciate the constructive insights, which have helped us further improve the clarity
> and presentation of the manuscript. We have carefully addressed every point and revised the paper
> accordingly, with all corresponding changes highlighted in blue.
> We hope that our clarifications resolve the reviewer’s remaining concerns,
> and we would be glad to provide any additional information if helpful.
>
> > **W1.1 & Q2:** The presentation of the operator $L$ and its components lacks precision. How exactly is $L_{physics}$ computed in practice?
>
> **A1:** $L_{physics}$ is not used as a feature input but is constructed directly as a sparse matrix operator. Specifically, $L_{physics}$ is a sparse matrix explicitly calculated based on the mesh geometry (see lines 261-267 of the paper), while $L_{neural}$ is a correction matrix composed of edge weights output by the GNN (see Eq. 10). The final operator is the element-wise sum of the two, $L = L_{physics} + L_{neural}$, which forms an $N \times N$ matrix. This synthetic operator is directly substituted into the Discrete Green's formula (Eq. 7) for state updates. Therefore, this is essentially an operator learning process that embeds physical priors, rather than a simple feature mapping.
>
> > **W1.2:** The interaction between $L_{neural}$ (the MPNN correction) and the source term $f$ is only superficially discussed.
>
> **A2:** The interaction between $L_{neural}$ and the source term $f$ is explicit and physical. $L_{neural}$ first participates in modifying the system's master operator $L$, thereby changing the discrete Green's function $\mathbf{G}(\Delta t) = (\mathbf{I} - \frac{\Delta t}{2}L)^{-1}$. Then, the source term $f$ acts on the system through matrix multiplication with this Green's function: $\mathbf{G}(\Delta t) \cdot \frac{\Delta t}{2}(f^k + f^{k+1})$. This implies that the neural component indirectly determines the source response by optimizing the system's propagator. The principle of this architecture design originates from Green's function theory (see Eq. 3). The update of state $u$ (Eq. 7) is the discretization of this theory on graphs, where the source term $f$ and physical state $u$ are decoupled, which is precisely the foundation for our generalization to unseen sources.
>
> > **W1.3 & Q1:** The neural component is relatively minimal. Does “w/o $L_{neural}$” correspond to solving the PDE purely via numerical methods?
>
> **A3:** We would like to clarify that the assumption that ``w/o $L_{\text{neural}}$'' corresponds to a purely numerical solver is a misunderstanding. Even without $L_{\text{neural}}$, the architecture still includes the Residual GNN in the prediction path (see lines 299-304), which contributes to learning data-driven corrections beyond the physics prior. The $L_{\text{neural}}$ module corrects operator discretization errors, while the Residual GNN captures residual dynamics during rollout.
>
> We present complete ablation experiments for both components in our paper, and further conducted ablation experiments in more scenarios. The results are shown in the table below. It shows that removing either neural component leads to a noticeable increase in prediction error, confirming that the model is not purely numerical and that the neural modules play an essential role in learning. Histograms of newly added ablation studies are also included in Appendix D.4.
>
> | Dataset           | Metric | DGNet    | w/o $L_{physics}$ | w/o $L_{neural}$ | w/o Residual GNN | w/o Green |
> | ----------------- | ------ | -------- | ----------------- | ---------------- | ---------------- | --------- |
> | Complex Obstacles | MSE    | 6.69e-05 | 8.12e-03          | 6.10e-04         | 1.43e-04         | 4.23e-02  |
> |                   | RNE    | 0.0211   | 0.5679            | 0.3198           | 0.1554           | 0.8057    |
> | Laser Heat        | MSE    | 1.76e+01 | 1.32e+03          | 5.93e+02         | 4.06e+01         | 6.12e+03  |
> |                   | RNE    | 0.0102   | 0.1539            | 0.0944           | 0.0693           | 0.1824    |
> | FitzHugh-Nagumo   | MSE    | 1.18e-07 | 7.24e-06          | 6.27e-07         | 4.64e-07         | 5.39e-05  |
> |                   | RNE    | 0.0102   | 0.1539            | 0.0944           | 0.0693           | 0.1824    |

---

> ### Author Response · Authors · 2025-11-25
>
> > **W1.4 & Q3 & Q5:** Concerns about robustness. The paper emphasizes strong generalization to unseen source terms, but given the small test sizes. Could larger-scale experiments be conducted to further validate generalization?
>
> **A4:** To evaluate robustness, we conducted extensive zero-shot stress tests by varying source bandwidth, amplitude, and the number of concurrent sources from $0.5\times$ to $3\times$ the levels seen during training. As shown in the table below, DGNet maintains stable accuracy across these unseen conditions, demonstrating strong robustness to changes in source behavior. Detailed settings are provided in Appendix F.4.
>
> |                              | 0.5 ×    | 1 ×      | 2 ×      | 3 ×      |
> | ---------------------------- | -------- | -------- | -------- | -------- |
> | bandwidth                    | 1.53e+01 | 1.76e+01 | 3.88e+01 | 7.36e+01 |
> | amplitude                    | 2.27e+01 | 1.76e+01 | 1.62e+01 | 1.95e+01 |
> | number of concurrent sources | 1.09e+01 | 1.76e+01 | 3.26e+01 | 4.40e+01 |
>
> To further validate the model's generalization ability in different scenarios, we now add a new light-driven chemical reaction experimental scenario. This scenario employs a richer combination of test trajectories (approximately 200) and is expanded to a larger scale (nearly 30,000 nodes). The experimental results are shown below, and detailed settings can be found in Appendix F.3.
>
> | Metric | DeepONet | MGN      | MP-PDE   | BENO     | PhyMPGN  | DGNet    |
> | ------ | -------- | -------- | -------- | -------- | -------- | -------- |
> | MSE    | 6.11e+03 | 1.93e+02 | 1.37e+01 | 6.04e+02 | 2.87e+02 | **3.97e+00** |
> | RNE    | 0.8832     | 0.1582   | 0.0422   | 0.2796   | 0.1921   | **0.0227** |
>
> > **W1.5:** Several mathematical derivations (Eqs. 5–7) presented without intuitive interpretation or discussion.
>
> **A5:** Eq. 5 is essentially the spatiotemporal discretization of the PDE, where time is discretized as $\Delta t$, and spatially, the PDE operator $\mathcal{L}_{\mathbf{x}}$ is discretized as matrix $L$. After derivation, we obtain Eq. 7, which is the system physical state update formula based on the superposition principle. The first term on the right corresponds to the natural evolution of the state at the previous time step, and the second term corresponds to the system's response to the source term. In this physical process, the $N \times N$ discrete Green's matrix $\mathbf{G}(\Delta t)$ serves as the propagation operator. It multiplies the modified state vector $\left( \mathbf{I} + \tfrac{\Delta t}{2} L \right) \mathbf{u}^k$ and the time-averaged source term vector $\tfrac{\Delta t}{2} \left( \mathbf{f}^k + \mathbf{f}^{k+1} \right)$ (both are $N \times 1$ column vectors), thereby propagating the PDE operator action and source term excitation to the entire spatial domain to obtain the system state at the next time step.
>
> > **Q4:** How does DGNet compare to PHYMPGN (ICLR 2025) under identical experimental settings? The differences in reported performance seem unexpectedly large.
>
> **A6:** Regarding PhyMPGN (ICLR 2025), we used the official open-source code from the paper. After reproduction, we found that its implementation has strong scenario specificity. The code is specifically customized for Navier-Stokes fluid equations (e.g., the network architecture mandates inputting fluid-specific physical parameters such as Reynolds number $Re$, and the physics encoding module hard-codes the velocity-pressure coupling of N-Stokes equations).
>
> However, our experimental scenario is completely different (for the seemingly most similar cylinder scenario, we study contaminant transport, solving for a scalar concentration field). This fundamental difference in physical governing equations means that PhyMPGN's original physical bias is unsuitable for our scenario. Despite our efforts to adjust its configuration to fit our experiments, its specially customized architecture still led to significant performance degradation in our setup.

---

> ### Author Response · Authors · 2025-11-27
>
> Dear Reviewer,
>
> Thank you again for the time you have spent evaluating our paper. As the deadline for the author–reviewer discussion is approaching, we would greatly appreciate any feedback you may be able to provide.
>
> We hope that our responses have addressed your concerns, and we would be glad to clarify anything further if needed.
>
> Thank you once again for your time and consideration.
>
> Best regards,
>
> The Authors

---

### Official Review · Reviewer_xgKd · 2025-11-04

**Soundness:** 3
**Presentation:** 3
**Contribution:** 3
**Rating:** 6
**Confidence:** 4

**Summary:**

This paper proposes DGNet, a neural solver for spatiotemporal PDEs that explicitly decouples the system evolution from the source response by turning Green’s function theory into a discrete, graph-based update rule. Starting from a Crank–Nicolson semi-discretization, the authors derive a one-step update that can be interpreted as applying a discrete Green operator to propagate the current state and accumulate the effect of the source over the time slab. Experiments span classical PDEs, transport on irregular domains, and generalization to unseen source terms. DGNet reports SOTA accuracy across tasks and especially large gains when tested on entirely novel sources; an ablation suggests the discrete Green solver is the key driver of performance.

**Strengths:**

- Solid derivation of a discrete Green update tied to an implicit Crank–Nicolson step, with practical “factorize once, solve many” sparse solves; the operator depends only on mesh and \Delta t, enabling reuse during rollout.

- Well-motivated hybrid operator that anchors learning with physics priors (Green–Gauss gradient, cotangent Laplacian) plus GNN corrections; clear architectural overview.

- Broad empirical coverage with consistent SOTA on both log-MSE and relative $\ell_2$ across diverse regimes; the table highlights DGNet edges on every scenario.

**Weaknesses:**

- Some implementation details (e.g., GPU sparse LU and custom adjoint) are in the appendix; a brief complexity/throughput table in the main text would help readers assess practicality across datasets.

- Uses standard message-passing blocks; novelty is concentrated in the Green discretization + operator split, not in GNN mechanics.

- Although there is an ablation, the residual GNN’s incremental benefit and the sensitivity to factorization accuracy (e.g., fill-in thresholds, preconditioners) could be quantified more rigorously beyond one scenario.

- While the CN-derived update is principled, the scope of PDE classes is primarily parabolic/weakly nonlinear; truly strongly nonlinear or hyperbolic systems are deferred to future work.

- Stability/consistency guarantees for the learned correction are not theoretically analyzed.

- Benchmarks are 2D; without a 3D case, claims about broad scientific impact are somewhat aspirational.

**Questions:**

1. Stability & constraints on $L_{\text{neural}}$.
Can you provide spectral/energy bounds or constraints on $L_{\text{neural}}$ (e.g., skew-symmetry, diagonally dominant correction, Lipschitz bounds) that preserve the A-stability properties of the CN-like update when combined with $L_{\text{physics}}$? A short lemma or empirical spectrum plots across datasets would strengthen confidence.

2. Ablations beyond one geometry.
Your ablation indicates “w/o Green” is worst on complex obstacles. Could you add two more scenarios (one classical PDE, one laser heat) to separate the contributions of $L_{\text{physics}}$, $L_{\text{neural}}$, and the residual GNN more generally? This would clarify the portability of each component.

3. Runtime, memory, and scaling.
Please include a main-text table: per-step wall-clock, memory footprint, and speedup vs. baselines at equal accuracy, and a mesh-scaling plot up to your largest meshes. What happens if the sparse LU factorization is replaced with ILU/AMG preconditioning for larger systems?

4. Robustness to source characteristics.
The laser task varies paths; could you report sensitivity to source bandwidth/amplitude and number of concurrent sources (e.g., 1 vs. 10 lasers), including failure modes where superposition might break due to strong nonlinearity?

---

> ### Author Response · Authors · 2025-11-25
>
> We sincerely thank the reviewer for the positive assessment and the detailed feedback.
> We truly appreciate the constructive insights, which have helped us further improve the clarity
> and presentation of the manuscript. We have carefully addressed every point and revised the paper
> accordingly, with all corresponding changes highlighted in blue.
> We hope that our clarifications resolve the reviewer’s remaining concerns,
> and we would be glad to provide any additional information if helpful.
>
> > **W1 & Q3:** Runtime, memory, and scaling. A brief complexity/throughput table in the main text would help readers assess practicality across datasets. Please give speedup vs. baselines at equal accuracy, and a mesh-scaling plot up to your largest meshes. What happens if the sparse LU factorization is replaced with ILU/AMG preconditioning for larger systems?
>
> **A1：**
>
> **Throughput table:** To demonstrate DGNet's practicality, we report the computational overhead and throughput in the table below (measured on a single RTX 4090). Both this table and the corresponding mesh scalability plot have been included in Appendix D.3 in the updated manuscript. As shown in the table, DGNet demonstrates robust inference scalability, with the per-step runtime increasing only marginally despite an order-of-magnitude growth in mesh resolution.
>
> | Dataset           | Nodes ($N$) | Pre-comp. Time (s) | Inference Time (ms/step) | Total Inference Time (s) | Throughput (steps/s) | Peak Memory (MB) |
> | ----------------- | ----------- | ------------------ | ------------------------ | ------------------------ | -------------------- | ---------------- |
> | Allen-Cahn       | 1,296       | 0.62        | 7.74              | 8.36                     | 129.17               | 152.60           |
> | Fisher-KPP       | 1,681       | 3.64        | 7.57              | 15.76                    | 131.99               | 227.97           |
> | FitzHugh-Nagumo  | 10,000      | 10.73       | 10.87             | 23.78                    | 91.98                | 3830.72          |
> | Cylinder         | 2,275       | 2.51        | 7.78              | 18.09                    | 128.38               | 368.81           |
> | Sediments        | 5,758       | 6.04        | 7.79              | 21.63                    | 128.30               | 1570.49          |
> | Complex Obstacles| 8,841       | 9.61        | 9.73              | 29.08                    | 102.70               | 3258.10          |
> | Laser Heat       | 6,072       | 1.93        | 7.73              | 2.86                     | 129.26               | 1570.65          |
>
> **Complexity Analysis:** Since the matrix $A = (I - \frac{\Delta t}{2}L)$ is highly sparse (non-zero only at mesh edges), we utilize `cupy.sparse.linalg.splu` for efficient factorization. This solver employs the Nested Dissection algorithm to minimize fill-ins, ensuring controlled complexity:
>
> - Space Complexity: $\mathcal{O}(N \log N)$, ensuring controllable GPU memory usage.
> - Pre-computation Complexity: $\mathcal{O}(N^{1.5})$, corresponding to the one-time sparse LU factorization.
>
> **Comparison of Inference Efficiency with Baselines:** The table below provides the single-trajectory inference time (in seconds) for DGNet and all baseline models on different datasets. This table has been included in Appendix D.3. As shown, DGNet exhibits inference time on the same scale as other baselines.
>
> |                   | DeepONet | MGN   | MP-PDE | BENO   | PhyMPGN | DGNet |
> | ----------------- | -------- | ----- | ------ | ------ | ------- | ----- |
> | Allen-Cahn        | 1.33     | 5.12  | 5.96   | 14.06  | 6.71    | 8.37  |
> | Fisher-KPP        | 2.77     | 12.74 | 19.20  | 75.92  | 17.40   | 15.76 |
> | FitzHugh-Nagumo   | 106.52   | 15.48 | 45.23  | 67.32  | 10.72   | 23.78 |
> | Cylinder          | 6.38     | 20.15 | 24.90  | 133.01 | 12.47   | 18.10 |
> | Sediments         | 58.94    | 21.47 | 31.25  | 101.10 | 14.59   | 21.63 |
> | Complex Obstacles | 141.92   | 23.59 | 47.36  | 108.96 | 20.31   | 29.08 |
> | Laser Heat        | 4.22     | 2.56  | 3.95   | 2.30   | 1.73    | 2.87  |
>
> Regarding the requested speedup at the same accuracy, since the baselines are all neural network methods, their inference speed depends on network architecture parameters rather than accuracy thresholds, so this metric cannot be directly calculated. Moreover, in our experiments, even when the baseline models converged, they could not reach the high accuracy level of DGNet. Therefore, we cannot provide the requested comparison.
>
> **Preprocessing with ILU/AMG:** We view this as a practical trade-off and a standard strategy for scaling DGNet to very large systems. Preconditioners such as ILU/AMG reduce the fill-in of exact LU decomposition and lower storage complexity to nearly linear $O(N)$, enabling scalability to larger meshes. The trade-off is that inference becomes iterative rather than direct, which increases single-step inference time accordingly.

---

> ### Author Response · Authors · 2025-11-25
>
> > **W2:** Novelty is concentrated in the Green discretization + operator split, not in GNN mechanics.
>
> **A2：** We fully agree that DGNet's core innovation lies in its Discrete Green's Function architecture and the embedding of physical priors, rather than the GNN modules themselves. This is because we believe that when faced with complex spatiotemporal PDEs, data-driven GNNs (even complex variants) often struggle to efficiently learn physical laws from limited data, resulting in data inefficiency and limited generalization. Therefore, we use standard GNNs only as adapters for processing graph data, focusing our core contribution on cleverly integrating powerful physical priors through the Discrete Green's Formula. This architectural innovation significantly improves data efficiency and solves the source term generalization problem. Furthermore, DGNet serves as a generic framework, capable of integrating with more expressive GNN backbones to further enhance performance.
>
> > **W3 & Q2:** Ablations beyond one geometry. The residual GNN’s incremental benefit and the sensitivity to factorization accuracy (e.g., fill-in thresholds, preconditioners) could be quantified more rigorously beyond one scenario.
>
> **A3：** We now add ablation studies under more scenarios, as shown in the table below. These results clearly demonstrate the incremental benefit yielded by the Residual GNN. Corresponding histograms have also been included in the appendix D.4.
>
> | Dataset         | Metric | DGNet    | w/o $L_{physics}$ | w/o $L_{neural}$ | w/o Residual GNN | w/o Green |
> | --------------- | ------ | -------- | ----------------- | ---------------- | ---------------- | --------- |
> | Laser Heat      | MSE    | 1.76e+01 | 1.32e+03          | 5.93e+02         | 4.06e+01         | 6.12e+03  |
> |                 | RNE    | 0.0102   | 0.1539            | 0.0944           | 0.0693           | 0.1824    |
> | FitzHugh-Nagumo | MSE    | 1.18e-07 | 7.24e-06          | 6.27e-07         | 4.64e-07         | 5.39e-05  |
> |                 | RNE    | 0.0102   | 0.1539            | 0.0944           | 0.0693           | 0.1824    |
>
> **Factorization accuracy:** We wish to clarify that we do not use iterative solvers such as ILU or AMG. Instead, we employ direct sparse LU factorization (a SuperLU-based direct solver via `cupy.sparse.linalg.splu`). Since the matrix $\mathbf{A} = (\mathbf{I} - \frac{\Delta t}{2}\mathbf{L})$ is strongly diagonally dominant, this factorization is highly stable and precise. Consequently, sensitivity to factorization parameters (e.g., fill-in thresholds or preconditioners) is not a concern in our current setup.
>
> > **W4:** The scope of PDE classes is primarily parabolic/weakly nonlinear; truly strongly nonlinear or hyperbolic systems are deferred to future work.
>
> **A4：**
>
> **Strong Nonlinearity:** First, we wish to clarify that while the evaluated equations (e.g., Allen-Cahn and FitzHugh-Nagumo) are mathematically semilinear parabolic, they exhibit strongly nonlinear dynamic characteristics in a physical sense. The high-order source terms in these systems are not merely small perturbations of linear solutions; rather, they dominate drastic transitions between different equilibrium points (e.g., phase transitions, wavefront propagation), which constitute strong nonlinearity in physics. We acknowledge that the phrasing "Strongly Nonlinear" in the "Future Work" section was imprecise; we specifically intended to refer to mathematically quasilinear PDEs (where coefficients of highest-order derivatives depend on $u$ or its lower-order derivatives). We have corrected this terminology in the updated manuscript.
>
> **Hyperbolic Systems:** While our current derivation primarily targets parabolic equations, the Discrete Green paradigm of DGNet can be extended to hyperbolic PDEs with minor modifications. By adapting the time integration scheme to use an Implicit Central Difference format for the second-order time derivative $\frac{u^{k+1} - 2u^k + u^{k-1}}{\Delta t^2}$, we derive the single-step update formula as follows:
>
> $$\mathbf{u}^{k+1} = \mathbf{G}(\Delta t) \left( 2\mathbf{I} + \frac{\Delta t^2}{2}\mathbf{L} \right) \mathbf{u}^k - \mathbf{G}(\Delta t)\mathbf{u}^{k-1} + \mathbf{G}(\Delta t)\frac{\Delta t^2}{2}(\mathbf{f}^{k} + \mathbf{f}^{k+1})$$
>
> Where $\mathbf{G}(\Delta t) = \left( \mathbf{I} - \frac{\Delta t^2}{2}\mathbf{L} \right)^{-1}$ is the discrete Green's operator for the hyperbolic system. This demonstrates that the core philosophy of DGNet is not limited to parabolic equations. We are actively working on further extending DGNet's application to hyperbolic systems.

---

> ### Author Response · Authors · 2025-11-25
>
> > **W5 & Q1:** Stability/consistency guarantees for the learned correction are not theoretically analyzed. A short lemma or empirical spectrum plots across datasets would strengthen confidence.
>
> **A5:** We thank the reviewer for the professional suggestions. We have added a stability and consistency analysis of the learned operator in Appendix E.
>
> Complementing this theoretical derivation, we have included empirical spectrum plots in Appendix D.5 as suggested. We compare the eigenvalue distributions of the physical operator $\mathbf{L}_{physics}$ and the total hybrid operator $\mathbf{L}$ across different datasets.
>
> To visualize the similarity between the learned and true operators, as well as the actual magnitude of the $\mathbf{L}_{neural}$ correction, please refer to Appendix D.1 (Figures 7 & 8) of the original manuscript.
>
> > **W6:** Benchmarks are 2D; without a 3D case, claims about broad scientific impact are somewhat aspirational.
>
> **A6:** To further validate the generality and scalability of our method, we have conducted transient heat conduction experiments in a 3D domain (containing approximately 6,000 nodes). As shown in the table below, DGNet maintains superior prediction accuracy in 3D geometries. For detailed experimental settings, please refer to Appendix F.1.
>
> | Metric | DeepONet | MGN      | MP-PDE   | BENO     | PhyMPGN  | DGNet        |
> | ------ | -------- | -------- | -------- | -------- | -------- | ------------ |
> | MSE    | 4.41e+03 | 4.60e+02 | 2.59e+04 | 5.28e+02 | 8.83e+03 | **4.42e+01** |
> | RNE    | 0.1630   | 0.1249   | 0.1993   | 0.0938   | 0.2352   | **0.0314**   |
>
> > **Q4:** Robustness to source characteristics. Could you report sensitivity to source bandwidth/amplitude and number of concurrent sources, including failure modes where superposition might break due to strong nonlinearity?
>
> **A7:** To evaluate robustness, we conducted extensive zero-shot stress tests by varying source bandwidth, amplitude, and the number of concurrent sources from $0.5\times$ to $3\times$ the levels seen during training. As shown in the table below, DGNet maintains stable accuracy across these unseen conditions, demonstrating strong robustness to changes in source behavior. Detailed settings are provided in Appendix F.4.
>
> |                              | 0.5 ×    | 1 ×      | 2 ×      | 3 ×      |
> | ---------------------------- | -------- | -------- | -------- | -------- |
> | bandwidth                    | 1.53e+01 | 1.76e+01 | 3.88e+01 | 7.36e+01 |
> | amplitude                    | 2.27e+01 | 1.76e+01 | 1.62e+01 | 1.95e+01 |
> | number of concurrent sources | 1.09e+01 | 1.76e+01 | 3.26e+01 | 4.40e+01 |
>
> To examine cases where the superposition principle may break, we further tested DGNet on a quasilinear phase-transition equation in which both the effective heat capacity $c_{eff}(T)$ and thermal conductivity $k(T)$ exhibit abrupt temperature-dependent jumps:
>
> $$\rho c_{eff}(T) \frac{\partial T}{\partial t} = \nabla \cdot (k(T) \nabla T) + Q(x,t).$$
>
> This induces strong nonlinearity that fundamentally violates linear superposition. As expected, DGNet’s accuracy degrades in this extreme regime. We include visualizations in Appendix F.5 to clearly delineate the applicability limits of our method.

---

> ### Author Response · Authors · 2025-11-27
>
> Dear Reviewer,
>
> Thank you again for the time you have spent evaluating our paper. As the deadline for the author–reviewer discussion is approaching, we would greatly appreciate any feedback you may be able to provide.
>
> We hope that our responses have addressed your concerns, and we would be glad to clarify anything further if needed.
>
> Thank you once again for your time and consideration.
>
> Best regards,
>
> The Authors

---

### Meta-Review · Area_Chair_kc7w · 2025-12-29

**Summary:**

I find this paper makes a good contribution to the neural PDE solver literature by addressing source-term generalization through a principled discrete Green's function formulation. The core insight—that treating source terms as first-class inputs rather than concatenated features enables out-of-distribution generalization—is well-motivated by classical PDE theory and demonstrated in experiments.

The reviewers raised concerns about presentation clarity, experimental scale, baseline fairness, and theoretical analysis. The authors provided a substantial rebuttal adding 3D experiments, large-scale benchmarks (~30k nodes), comparisons with recent strong baselines (SINGER, AMG-GNN, Transolver), detailed cost tables, stability analysis, and stress tests varying source characteristics. I find these additions address the major criticisms. The method's main limitation to semilinear PDEs is appropriately acknowledged. While the problem sizes remain modest by industrial standards, they are consistent with comparable work in this area. The experimental improvements, particularly the laser heat experiments showing robust generalization to unseen source patterns while baselines collapse, demonstrate the method's value.

**Reviewer Concerns:**

Reviewer xgKd requested 3D experiments, runtime comparisons, stability analysis, and extended ablations. The rebuttal provided all of these: a 3D heat conduction benchmark, inference time tables showing DGNet is competitive with baselines, eigenvalue spectrum plots confirming stability, and ablations across FitzHugh-Nagumo and Laser Heat scenarios. I consider these concerns mostly resolved.

Reviewer HxRT questioned how L_physics is computed and whether the neural component is meaningful. The authors clarified that L_physics is constructed as a sparse matrix operator (not feature input) and that removing either L_neural or the residual GNN increases prediction error across scenarios. The concern about small test sets is partially mitigated by the new 200-trajectory light-driven reaction experiment. I consider the clarity concern addressed; the statistical concern remains minor given the consistent performance patterns.

Reviewer dKtN asked about extension to 3D, comparison with classical solvers, and notation clarity. The rebuttal added 3D experiments, speed-accuracy tradeoff plots versus FEM, and improved notation with schematic figures. I consider these concerns resolved.

Reviewer pbbn raised concerns about pre-factorization costs, problem scale, and baseline selection. The cost tables show pre-factorization takes only seconds and inference time is comparable to baselines. The new 30k-node experiment and 3D benchmark address scale concerns. The addition of SINGER, AMG-GNN, and Transolver as baselines addresses the missing comparisons. I consider the fairness concerns resolved; the scope limitation to semilinear PDEs remains but is acknowledged.

**Reviewer Scores:**

Reviewer xgKd: 6 → 7. Concerns about 3D, runtime, stability, and ablations were fully addressed.

Reviewer HxRT: 4 → 5. Clarity and ablation concerns addressed. Small test set concern partially mitigated by new experiments.

Reviewer dKtN: 6 → 7. Notation, 3D, and numerical comparison concerns addressed. Likely satisfied.

Reviewer pbbn: 4 → 5. Cost tables, 3D experiments, large-scale benchmarks, and new baselines address main concerns.

---

### Decision · Program_Chairs · 2026-01-26

Accept (Poster)